# *UniGen*: Enhanced Training & Test-Time Strategies for Unified Multimodal Understanding and Generation

**Rui Tian**[1][2][∘]**, Mingfei Gao**[2][∘]**, Mingze Xu**[2][∘]**,**
**Jiaming Hu**[2]**, Jiasen Lu**[2]**, Zuxuan Wu**[1][†]**, Yinfei Yang**[2]**, Afshin Dehghan**[2][†]
[1]Institute of Trustworthy Embodied AI, Fudan University    [2]Apple
{mgao22,mingze_xu2,adehghan}@apple.com, {rtian23,zxwu}@fudan.edu.cn,
[∘]First authors; [†]Corresponding authors

## Abstract

We introduce ***UniGen***, a unified multimodal large language model (MLLM) capable of image understanding and generation. We study the full training pipeline of *UniGen* from a data-centric perspective, including multi-stage pre-training, supervised fine-tuning, and direct preference optimization. More importantly, we propose a new ***Chain-of-Thought Verification (CoT-V)*** strategy for test-time scaling, which significantly boosts *UniGen*'s image generation quality using a simple *Best-of-N* test-time strategy. Specifically, *CoT-V* enables *UniGen* to act as both image generator and verifier at test time, assessing the semantic alignment between a text prompt and its generated image in a step-by-step CoT manner. Trained entirely on open-source datasets across all stages, *UniGen* achieves state-of-the-art performance on a range of image understanding and generation benchmarks, with a final score of $0.78$ on GENEVAL and $85.19$ on DPG-BENCH. Through extensive ablation studies, our work provides actionable insights and addresses key challenges in the full life cycle of building unified MLLMs, contributing meaningful directions to future research. Code is available at https://github.com/apple/ml-unigen.

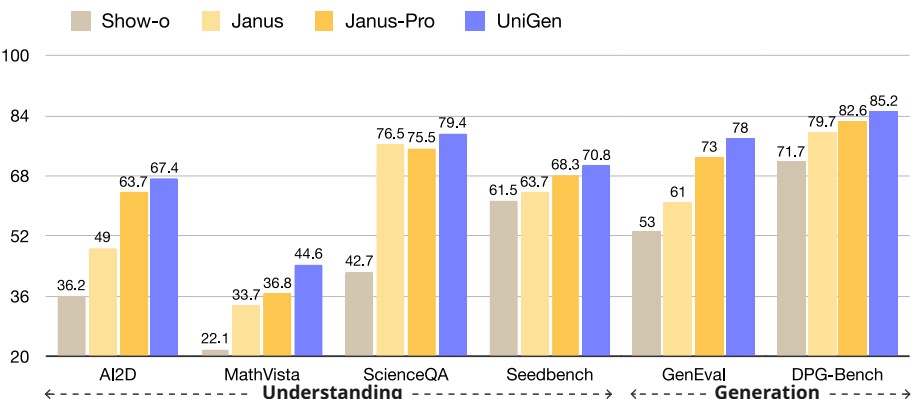

Figure 1: **Comparison against state-of-the-art unified MLLMs.** *UniGen-1.5B* outperforms Show-o-1.3B, Janus-1.3B and Janus-Pro-1.5B across understanding and generation benchmarks.

## 1   Introduction

Unifying understanding and generation within a single framework represents a key step toward general-purpose artificial intelligence models [53]. Pioneering work [9, 16, 67, 83, 84, 87, 96] has made encouraging progress but relies on distinct training recipes and internal datasets. More

39th Conference on Neural Information Processing Systems (NeurIPS 2025).

importantly, they have yet to demonstrate good practice in wisely collaborating these two capabilities within a unified architecture to achieve substantial performance gains. We advance the development of unified multimodal large language models (MLLMs) by carefully studying the impact of their training recipes across different stages and proposing optimizations to improve both image understanding and generation. We further explore leveraging test-time interaction between understanding and generation tasks, selecting images with higher quality by using our unified MLLM as the self-verifier.

Specifically, we introduce *UniGen*, a unified MLLM for image understanding and generation. To shed light on the impact of different training stages, we walk through the entire life cycle of the model development, including multi-stage pretraining, supervised fine-tuning [40, 54, 56], and direct preference optimization [60, 78]. We ablate the impact of each training stage and their design choices from a data-centric perspective, and draw insightful lessons for building advanced unified MLLMs. Unlike state-of-the-art models [9, 47, 87, 84] that rely on large-scale internal datasets, we curate new data mixtures across training stages by using only open-source images. We show that models trained on publicly available data can also achieve competitive results.

To further enhance image generation quality, we propose a new ***Chain-of-Thought Verification*** (***CoT-V***) strategy for test-time scaling. The key idea is to leverage *UniGen*'s inherent understanding ability as a self-verifier to assess the quality of its own generated images. Specifically, during inference, *UniGen* produces $N$ images for a given text prompt, while *CoT-V* progressively evaluates semantic coherence between each image-text pair and selects the best. With only lightweight fine-tuning (*e.g.*, 500 training steps), *UniGen* is able to achieve the reasoning capability, thinking step-by-step to verify each atomic fact according to the prompt and each generated image. Importantly, this CoT verification seamlessly enhances *UniGen*'s image generation quality while preserving its general understanding performance. In this way, we collaborate the understanding and generation capabilities within a unified MLLM, substantially boosting the text-to-image generation quality using a simple *Best-of-N* strategy [79, 106] and self-verification [82, 7, 24]. Our experiments show that *UniGen*'s performance is consistently improved across various image generation benchmarks.

We evaluate *UniGen* on various understanding and generation tasks, as shown in Fig. 1. For image understanding, *UniGen* outperforms comparable unified MLLMs (*e.g.* Show-o [87] and Janus-Pro [9]) across benchmarks and even ties with some strong understanding specialist models, such as LLaVA-OV [32] and MM1.5 [102], as displayed in Table 1. For text-to-image generation, *UniGen* obtains 0.78 on GENEVAL and 85.19 on DPG-BENCH using only open-source data, surpassing state-of-the-art unified MLLMs [87, 84, 9] by a clear margin.

## 2 Related Work

**Multimodal Large Language Models (MLLMs)** have advanced significantly in image [1, 11, 40, 41, 49, 77, 106] and video understanding [42, 103, 89, 90, 101, 108]. Their architecture typically consists of a vision encoder [59, 100, 72] to extract visual features, a projector [34, 2] to align image-text embeddings, and a large language model (LLM) [1, 71, 92, 10] to generate responses. Early work focuses on pre-training using large-scale vision-language corpus [48, 69], then moves to carefully curated instructional datasets for supervised fine-tuning [32, 102] and reinforcement learning [27, 60]. Recently, enabling MLLMs to output explicit reasoning trajectories has become a promising research direction [52, 19, 68]. They explore strategies, such as chain-of-thought (CoT) prompting [13, 95], reinforcement learning [66, 88], and test-time scaling [106, 79] to enhance the visual reasoning capabilities of MLLMs.

**Unified Understanding and Generation** aims to combine visual understanding and generation within a single MLLM framework [51, 67, 58, 81, 44, 43, 8, 12, 35]. This is often achieved by jointly optimizing LLMs with multimodal objectives and generation-specific losses, such as autoregressive decoding [84], diffusion [105], flow-matching [47], and masked image prediction [94, 87]. Visual tokenizers [14, 45, 50, 73, 75, 98, 104] are critical for enabling both semantic understanding and high-fidelity generation. Recent efforts explore both decoupled encoders [70, 84] and unified tokenizers [29, 58, 85] for better task balancing. Integrating CoT into visual generation emerges as a promising strategy. PARM [20] scales test-time computation by introducing a new verification process. MINT [81], ImageGen-CoT [38], and Got [15] leverage multimodal reasoning to perform prompt planning, generation, reflection, and refinement. Despite these advances, using chain-of-thought for unified understanding and generation remains underexplored. In this work, *UniGen* incorporates a CoT-based self-verification strategy via Best-of-N selection during test-time scaling, which leads to substantial improvements in image generation performance.

# 3 Recipe for Building *UniGen*

## 3.1 Architecture

As shown in Fig. 2, we unify the image understanding and generation tasks into a pretrained LLM. Motivated by prior work [84], we separate visual encoding for understanding and generation into continuous and discrete embedding spaces, respectively.

**For image understanding**, we follow the LLaVA [40] workflow and adopt the next-token prediction paradigm. Given an input image $X^U$, the understanding encoder $\mathbf{Enc}^U$ (*e.g.*, SigLIP [100]) extracts its feature as a vector of continuous tokens $\mathcal{X}^U = \mathbf{Enc}^U(X^U)$. The projector $\mathbf{P}^U$ aligns the image and text embeddings into the same space, then the embeddings are fed into

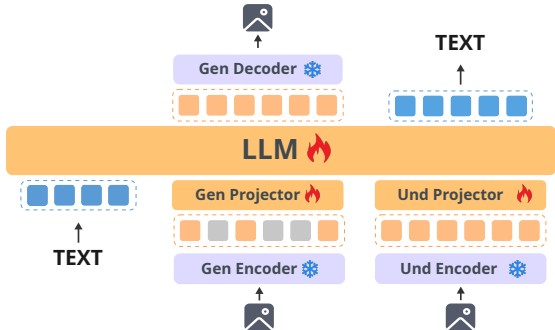

Figure 2: **The architecture of *UniGen***, which is based on an autoregressive LLM and decoupled vision encoders for image understanding and generation tasks.

LLM as inputs. We compute the understanding loss using the vanilla autoregressive training objective $\mathcal{L}_{und}$. To preserve the LLM's language modeling capability, we also train *UniGen* with text-only data and backpropagate the corresponding loss $\mathcal{L}_{text}$.

**For text-to-image generation**, we employ the masked token prediction [5] as our training objectives. Unlike the autoregressive decoding for text tokens, this paradigm enables models to generate multiple image tokens in parallel, significantly accelerating the generation process. *During training*, for each image $X^G \in \mathbb{R}^{H \times W}$, the generation encoder $\mathbf{Enc}^G$ (*e.g.*, MAGVIT-v2 [97]) tokenizes it into a sequence of discrete tokens $\mathcal{X}^G$ of length $N = H/d_s \cdot W/d_s$, where $d_s$ refers to the spatial downsampling factor of $\mathbf{Enc}^G$. Then, given a masking ratio $\eta$ according to the scheduling function $\gamma(\cdot)$, we randomly sample a binary mask $\mathcal{M}(\eta) = [m_0, \cdots, m_{N-1}]$, where $\eta * N$ positions are uniformly set to 1 and others are set to 0. For each position $i$ where $m_i$ equals to 1, we replace its corresponding discrete image token $\mathcal{X}_i^G$ with a special mask token $[\text{MASK}]$ to form the final input image sequence. Finally, we prepend the textual tokens (*e.g.*, image classes or captions) with the masked sequence $\mathcal{X}^{\mathcal{M}}$. *During inference*, the image generation starts with all masked tokens $\mathcal{X}^{\mathcal{M}} = [[\text{MASK}], \cdots, [\text{MASK}]]$, and gradually fills up the latent representation with scattered predictions in parallel.

## 3.2 Pre-Training (PT)

The goal of pre-training is to develop *UniGen*'s visual generation capability while preserving its potential for multimodal understanding. Thus, we only optimize the generation projector and the LLM with other parameters frozen. We also include image-to-text and text-only pre-training to keep *UniGen*'s language modeling capability. To encourage a better alignment between discrete image tokens and the text, we directly use the generation encoder for understanding tasks *but only* in this stage. We empirically find that this design can significantly improve the image generation performance. Specifically, we employ an "easy-to-difficult" strategy through a two-stage process.

**Pre-training Data.** We generate fine-grained captions for images from ImageNet [62] , CC-3M [63], CC-12M [6] and SAM-11M [31] dataset using Qwen2.5-VL-7B [3] to form a 40M image-text pair corpus. For text-only pre-training, we use RefinedWeb [55].

**PT-1 Stage** seeks to align the image and text embeddings and predict the distribution of basic visual concepts. Similar to prior works [84], we employ ImageNet for generation pre-training warmup and leverage the full 40M image-text pairs for the understanding task. However, we propose that *using image captions, rather than image categories, for text-to-image generation leads to better convergence*.

**PT-2 Stage** further facilitates *UniGen* to generalize to wider visual generation capabilities. We expand the text-to-image training dataset to the full 40M image-text pairs, while using the same image-to-text and text-only ones. We argue that *training data with a richer distribution enables more accurate control over generation patterns*. We name the model trained in this stage as ***UniGen-PT***.

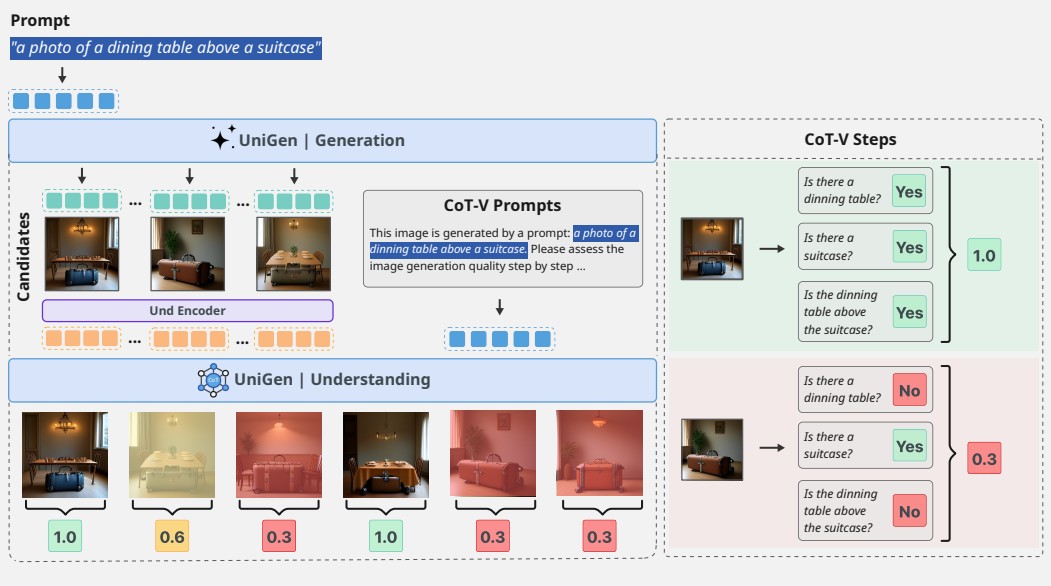

Figure 3: **The workflow of *UniGen* using test-time scaling and *CoT-V*. Left:** Illustration of *Best-of-N* selection with *CoT-V* with $N = 6$. **Right:** Visualization of the step-by-step reasoning process in *CoT-V* for computing the final quality score.

## 3.3 Supervised Fine-Tuning (SFT)

In the SFT stage, *UniGen* is jointly trained on the image understanding and generation tasks. We fine-tune the generation projectors, understanding projectors, and the LLM, while still keeping the vision encoders frozen. **For image understanding**, we notice that the knowledge-centric understanding is limited during pre-training stages. To enhance related capabilities, we adopt the strong image mixture from SlowFast-LLaVA-1.5 [90], which was carefully curated from open-source datasets with 4.67M multimodal VQA samples. **For image generation**, prior work [9] uses high-quality synthetic data that can enable fast and robust training convergence. We share this observation by using the JourneyDB [64] and text-2-image-2M [28] to improve the aesthetic quality of our generated images. We name the model trained in this stage as ***UniGen-SFT***.

## 3.4 Direct Preference Optimization (DPO)

We further enhance *UniGen* by aligning its outputs with human preference through DPO. We first discuss how we construct our synthetic preference dataset, then describe our DPO algorithm.

**Preference Dataset.** We leverage *UniGen-SFT* to generate the images for our preference dataset. For a given prompt, 20 images are generated. A preferred and rejected sample pair is constructed by evaluating the coherence between each image and the prompt. To improve the data robustness, we collect 6k short prompts from PARM [20], 6k medium-length prompts from T2I-Comp [25] training set, and 6k long prompts from re-annotated SA1B to generate training image candidates.

For short prompts, we use the GENEVAL metrics to evaluate the generation quality. For prompts of medium or long lengths, we decompose each prompt into fine-grained visual questions with Qwen2.5-7B. Then, we assess image-prompt consistency by feeding Qwen2.5VL-7B with the image-question pair. An output "yes" indicates the image aligns with the description, and "no" otherwise. The final consistency score $\mathcal{S}$ is averaged from these answers. For each prompt, we sample one highest-scored example as the preferred image and the lowest one as the rejected image. Prompts with no clear preference are filtered out. Finally, we obtain around 13k triplets for training.

**DPO Training.** We adopt the vanilla DPO training loss and freeze the understanding encoder and projector in this stage. The training ends in one epoch with a batch size of 64 and a learning rate of $1e^{-5}$. We empirically find that *this DPO training does not impair UniGen's understanding performance.* We name the model trained in this stage as ***UniGen-DPO***.

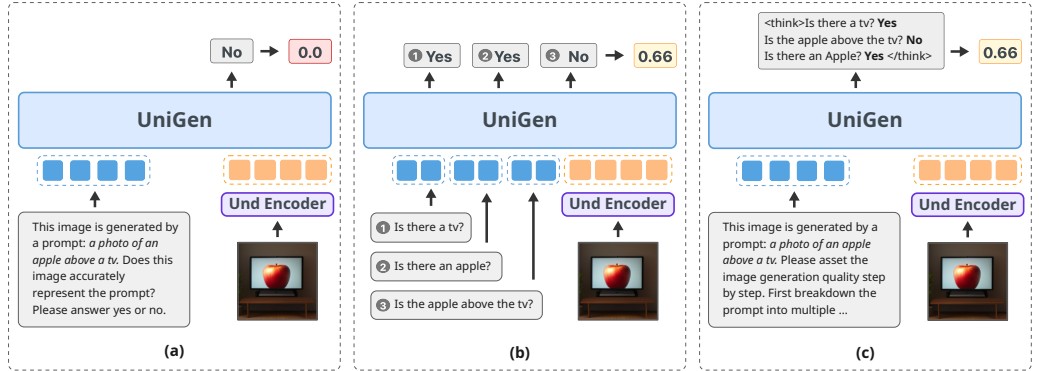

Figure 4: **An example of using different image verification methods**: **(a)** Outcome Verification, **(b)** Rule-based Verification and **(c)** Chain-of-Thought Verification.

### 3.5 Test-Time Scaling

Recent studies have shown the effectiveness of test-time scaling on improving both image understanding [79, 106] and generation [20, 86]. We employ the Best-of-N evaluation strategy and leverage *UniGen*'s understanding ability to conduct self-critique for image generation verification. The general workflow is illustrated in Fig. 3. *First*, *UniGen* generates $N$ candidate images for a given prompt. *Second*, we input each generated image along with its prompt into *UniGen*, which evaluates the alignment between the image and its textual description and outputs a quality score $\mathcal{S}$. *Third*, we return the image with the highest score. We propose three verification methods as shown in Fig. 4.

- **Outcome Verification (OV)** simply prompts *UniGen* to directly judge the coherence of the input prompt and each image candidate, giving a binary score (*i.e.* "yes" for a good match and "no" for a failure generation). We randomly select one if there are candidates with the same score.

- **Rule-based Verification (RV)** breaks down each prompt into several atomic questions based on pre-defined rules, then sequentially feeds them with the generated image into *UniGen* for quality verification. The results of all sub-questions are averaged as the final quality score.

- **Chain-of-Thought Verification (CoT-V)** instructs the model to think step-by-step and verifies each atomic fact according to the prompt and each generated image, following the CoT format: `<think_start>`$Q_1$? $A_1$; $\cdots$ $Q_n$? $A_n$;`<think_end>`. We compute the final quality score $\mathcal{S}$ by parsing the CoT outputs. Specifically, given a text prompt $T$ and a generated image $I$, *CoT-V* produces a list of visual questions $Q = \{Q_1, \cdots, Q_n\}$ and their corresponding answers $A = \{A_1, \cdots, A_n\}$. The final score $\mathcal{S}$ is computed by averaging the scores in the answer list.

OV relies on *UniGen*'s pattern-matching capabilities without intermediate reasoning. RV incorporates a rule-driven reasoning process into test-time scaling. Although effective on well-structured prompts, RV struggles with free-form or complex instructions, such as those in DPG-Bench [23]. *CoT-V* leverages the strengths of both approaches, enabling reasoning-driven image verification without the need for manual prompt decomposition. Thus, we use *CoT-V* as our default verification method.

#### 3.5.1 *CoT-V* Post-Training

*UniGen* has not been precisely trained to generate CoT responses. Here we introduce a lightweight post-training strategy upon *UniGen-DPO*, equipping it with the ability of CoT-based verification.

**Data.** To construct the *CoT-V* post-training data, we reuse the image-text pairs collected during the DPO stage (Sec. 3.4). For prompts sourced from PARM, we extract the question-answer pairs via rule-based matching, since they are built upon a clear structure [17]. For prompts from T2I-Comp that are more complicated, we first guide Qwen2.5-7B [91] to generate a series of atomic questions, then query Qwen2.5-7B-VL with each image-question pair to obtain their binary pseudo labels. We exclude the prompts from SA-1B due to the lower quality of the decomposed visual questions. We empirically find that most of the decomposed questions do not fully cover the visual concepts of the original caption. We totally sample 20K image-question-answer triplets from both prompt sources.

**Training.** We format the above 20K training pairs as instruction-following conversations, and feed them into *UniGen-DPO* for supervised fine-tuning. In this stage, we only optimize the understanding

Table 1: **Comparison with state-of-the-art models on image understanding benchmarks.**
*denotes reproduced results and RW-QA denotes RealWorld-QA.

| Model | #Params | Res. | AI2D | GQA | POPE | MMMU | MathVista | RW-QA | ScienceQA | Seedbench |
|---|---|---|---|---|---|---|---|---|---|---|
| | | | *Understanding MLLMs* | | | | | | | |
| LLaVA-OV [32] | 0.5B | AnyRes | 57.1 | - | - | 31.4 | 34.8 | 55.6 | 67.2 | 65.5 |
| MM1.5 [102] | 1B | AnyRes | 59.3 | - | 88.1 | 35.8 | 37.2 | 53.3 | 82.1 | 70.2 |
| LLaVA 1.5 [39] | 7B | 336 | 55.1* | 62.0 | 86.1 | 36.3* | 26.7* | 55.8* | 66.8 | 66.1 |
| | | | *Unified MLLMs* | | | | | | | |
| Show-o [87] | 1.3B | 336 | 36.2* | 61.0* | 84.5 | 27.4 | 22.1* | 48.5* | 42.7* | 61.5* |
| Janus [84] | 1.3B | 384 | 49.0* | 59.1 | 87.0 | 30.5 | 33.7* | 48.4* | 76.5* | 63.7 |
| Janus-Pro [9] | 1.5B | 384 | 63.7* | 59.3 | 86.2 | **36.3** | 36.8* | 51.1* | 75.5* | 68.3 |
| Vila-U [85] | 7B | 384 | - | 60.8 | 85.8 | - | - | - | - | 56.3 |
| MMAR [93] | 7B | 256 | - | - | 83.0 | - | - | - | - | 64.5 |
| UniToken [29] | 7B | 384 | **68.7** | - | - | 32.8 | 38.5 | - | - | 69.9 |
| *UniGen* | 1.5B | 384 | 67.4 | **62.3** | **87.8** | 32.3 | **44.6** | **56.7** | **79.4** | **70.8** |

projector and the LLM. To ensure not impairing *UniGen*'s general understanding capabilities, we fine-tune *UniGen* on this *CoT-V* dataset for only 500 steps using a small learning rate of $1 \times 10^{-5}$. The model trained after this stage is our final model, and we name it as **UniGen**.

## 4 Experiments

### 4.1 Implementation Details

We use 32 H100-80G GPUs for pre-training stages and 8 H100-80G GPUs for the others. *UniGen* is built upon the pre-trained Qwen2.5-1.5B [91]. We adopt MAGVITv2 from Show-o [87] as our discrete visual encoder with input resolution of $256 \times 256$ and SigLIP [100] as our continuous visual encoder. As discussed in Sec. 3.1, we use MAGVITv2 for both understanding and generation in PT-1 and PT-2, and keep using SigLIP as the understanding encoder after SFT.

**Training.** We follow Show-o [87] to use a bidirectional attention mask within image tokens, but keep the causality within text tokens and between multimodal tokens. Detailed hyperparameters for each training stage are described in Appendix Table 17 with more details in Appendix Sec. E.0.2.

**Inference and Evaluation.** We follow the common practice of image generation to use classifier-free guidance [22] and set the scale to 5.0. In addition, we follow MaskGIT [5] to adopt the cosine masking scheduler in inference and set the default number of steps to $T = 50$. We use MAGVITv2 decoder to project the visual tokens back to the pixel space. For test-time scaling with *CoT-V*, we generate $N = 20$ image candidates per text prompt and select top-K ($K = 5$) out of them, sending for evaluation on GENEVAL and DPG-BENCH.

### 4.2 Main Results

We report the performance of *UniGen* on various benchmarks (details are discussed in Appendix Sec. A) and show qualitative results in Fig 5. We mainly compare *UniGen* with state-of-the-art unified LLMs in Table 1 and 2, but also reference strong specialist models to understand our position in the whole picture of MLLMs. Here we highlight the following observation.

**First, *UniGen* achieves state-of-the-art results across understanding benchmarks compared to existing unified MLLMs.** Specifically, *UniGen* outperforms Janus-Pro on RealWorld-QA, AI2D and MathVista by $+5.6\%$, $+3.7\%$, and $+7.8\%$, respectively. We believe our improvements are mainly driven by using *(i)* the decoupled generation and understanding encoders and *(ii)* the stronger SFT data mixture. Notably, *UniGen* is even comparable with some strong understanding-only MLLMs, such as LLaVA-OV-0.5B and MM1.5-1B, even though they use much higher input resolutions.

**Second, *UniGen* significantly outperforms existing unified MLLMs and strong generation-only models on text-to-image benchmarks.** Using GENEVAL in Table 2 as an example, *UniGen* achieves the overall score of $0.78$, significantly outperforming Janus-Pro by $0.05$. Besides, our model demonstrates an overwhelming advantage on the "Counting" task by $+0.27$ higher than Janus-Pro. *UniGen* even beats a range of superior generation-only models (*e.g.*, outperforming DALLE-2, and Emu3 by $+0.26$, and $+0.24$, respectively), even though they are with much larger model sizes. Similarly, *UniGen* outperforms existing models by a clear margin on DPG-BENCH as shown in Table 2, outperforming Show-o and Janus-Pro by $+13.49$ and $+2.56$, respectively.

Table 2: **Comparison with state-of-the-art models on GENEVAL and DPG-BENCH benchmark.**

| Model | # Params | GenEval↑ | | | | | DPG-Bench↑ | | |
|---|---|---|---|---|---|---|---|---|---|
| | | Two Obj. | Counting | Position | Color Attri. | Overall | Global | Relation | Overall |
| *Text-to-Image Generation Models* | | | | | | | | | |
| DALLE-2 [61] | 6.5B | 0.66 | 0.49 | 0.10 | 0.19 | 0.52 | - | - | - |
| DALLE-3[4] | - | 0.87 | 0.47 | 0.43 | 0.45 | 0.67 | 90.97 | 90.58 | 83.50 |
| Emu3 [80] | 8B | 0.71 | 0.34 | 0.17 | 0.21 | 0.54 | 85.21 | 90.22 | 80.60 |
| SDXL [57] | 2.6B | 0.74 | 0.39 | 0.15 | 0.23 | 0.55 | 83.27 | 86.76 | 74.65 |
| SimpleAR [76] | 1.5B | 0.90 | - | 0.28 | 0.45 | 0.63 | 87.97 | 88.33 | 81.97 |
| Infinity [21] | 2B | 0.85 | - | 0.49 | 0.57 | 0.73 | 93.11 | 90.76 | 83.46 |
| *Unified MLLMs* | | | | | | | | | |
| Show-o [87] | 1.3B | 0.52 | 0.49 | 0.11 | 0.28 | 0.53 | 80.39* | 83.36* | 71.70* |
| Janus [84] | 1.3B | 0.68 | 0.30 | 0.46 | 0.42 | 0.61 | 82.33 | 85.46 | 79.68 |
| Janus-Pro [9] | 1.5B | 0.82 | 0.51 | 0.65 | 0.56 | 0.73 | 87.58 | 88.98 | 82.63 |
| ILLUME [74] | 7B | 0.86 | 0.45 | 0.39 | 0.28 | 0.61 | - | - | - |
| UniToken [29] | 7B | 0.80 | 0.35 | 0.38 | 0.39 | 0.63 | - | - | - |
| VARGPT-v1.1 [107] | 9B | 0.53 | 0.48 | 0.13 | 0.21 | 0.53 | 84.83 | 88.13 | 78.59 |
| TokenFlow-XL [58] | 13B | 0.72 | 0.45 | 0.45 | 0.42 | 0.63 | 78.72 | 85.22 | 73.38 |
| *UniGen* | 1.5B | 0.92 | 0.68 | 0.48 | 0.52 | **0.74** | 91.53 | 91.09 | **84.89** |
| *UniGen* + CoT-V | 1.5B | 0.94 | 0.78 | 0.57 | 0.54 | **0.78** | 91.95 | 92.04 | **85.19** |

Table 3: **Ablation of different stages of our model on image understanding benchmarks.**

| Model | Stage | GenEval | DPG-Bench | AI2D | GQA | POPE | MMMU | MathVista | RW-QA | ScienceQA | Seedbench |
|---|---|---|---|---|---|---|---|---|---|---|---|
| | PT-1 | 0.53 | 78.14 | - | - | - | - | - | - | - | - |
| | PT-2 | 0.55 | 80.71 | - | - | - | - | - | - | - | - |
| *UniGen* | SFT | 0.63 | 82.75 | 68.0 | 62.5 | 87.4 | 32.4 | 45.2 | 58.6 | 79.7 | 71.1 |
| | DPO | 0.73 | 84.89 | 67.9 | 62.4 | 88.0 | 32.9 | 45.0 | 59.0 | 79.5 | 71.0 |
| | *CoT-V* | 0.78 | 85.19 | 67.4 | 62.3 | 87.8 | 32.3 | 44.6 | 56.7 | 79.4 | 70.8 |

## 4.3 Ablation Studies

### 4.3.1 Impact of Different Training Stages

We examine our training pipeline by showing the understanding and generation performance after each stage in Table 3. Here we highlight some key observations.

**First, *UniGen* demonstrates consistent improvements in generation performance across different training stages**, as indicated by the increasing numbers of GENEVAL and DPG-BENCH. The pre-training stages aim to warm up the generation capability of *UniGen*. The SFT boosts the GENEVAL and DPG-BENCH by using high-quality generation datasets. With the effectiveness of our preference data, the DPO stage significantly improves GENEVAL and DPG-BENCH to $0.73$ ($+0.10$) and $84.89$ ($+2.14$), respectively. *CoT-V* further enhances the scores to $0.78$ ($+0.05$) and $85.19$ ($+0.3$) via test-time scaling.

**Second, *UniGen*'s strong understanding capability is stimulated in the SFT stage and can be maintained in the following stages.** The SFT stage promotes the instruction following capability of *UniGen* that leads to strong performance on understanding benchmarks. In the DPO stage, *UniGen* successfully maintains the strong understanding capability. *CoT-V* contains an additional lightweight fine-tuning to encourage the CoT verification during test-time scaling. The results show that it does not sacrifice the general understanding capability, except for a slight regression on RealWorld-QA. We attribute this regression to the distribution gap between *CoT-V*'s synthetic training data and the real-world images in RealWorld-QA.

### 4.3.2 Ablation of *CoT-V*

Here we evaluate different verification methods discussed in Sec. 3.5 with the following highlights.

**First, CoT verification achieves the best performance and prompting *UniGen*'s thinking process is important.** As shown in Table 4, using *Outcome verification* shows no improvement, while using CoT thinking obtains a significant boost of generation performance on both GENEVAL and DPG-BENCH. We also observe that *Rule-based verification* is also effective, leading to $0.75$ on GENEVAL. However, it is not general enough to be used on free-form prompts. Comparing the results from *CoT Verification* and *Rule-based Verification*, we can see that prompting the model itself to think is beneficial for more reliable critique.

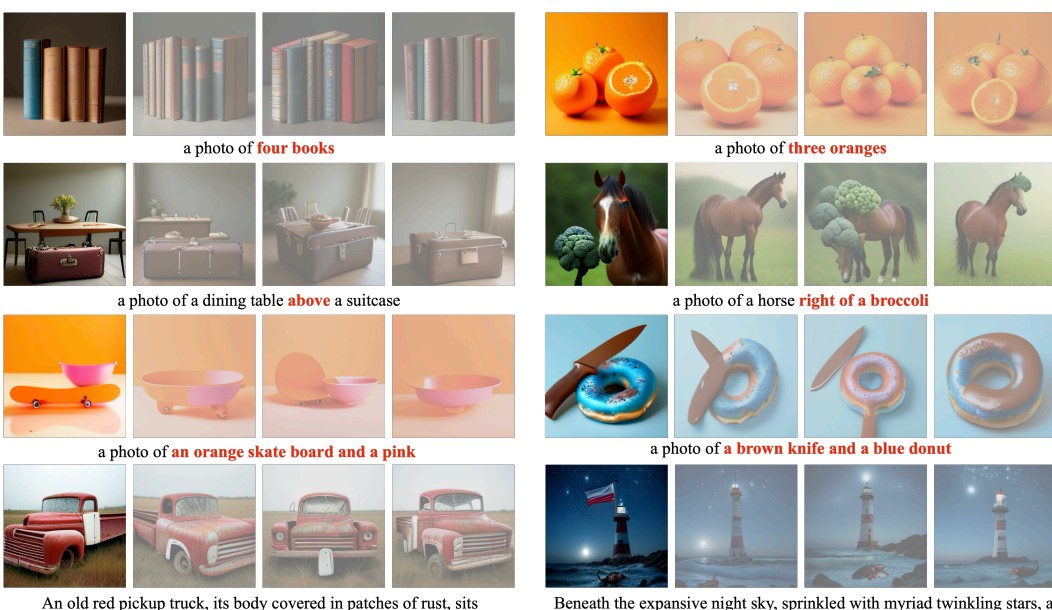

a photo of **four books**

a photo of **three oranges**

a photo of a dining table **above** a suitcase

a photo of a horse **right of a broccoli**

a photo of **an orange skate board and a pink**

a photo of **a brown knife and a blue donut**

An old red pickup truck, its body covered in patches of rust, sits abandoned in an open field. The truck's **white doors** stand in stark contrast to the faded red paint, and the windshield is shattered, with spiderweb cracks running across the glass. The vehicle's bed is empty, and the tires are worn, hinting at many years of service and neglect.

Beneath the expansive night sky, sprinkled with myriad twinkling stars, a **flag adorning the pinnacle of a towering lighthouse** drifts gently in the evening breeze, its hues subdued by the soft silver glow of the moon… At the water's edge, the rough texture of the rocks is intermittently made visible… out of the embrace of the ocean.

Figure 5: **Visual examples of *UniGen*'s results using *CoT-V*.** The first three rows show examples for counting, position, and color attribute, respectively, and the last row shows images generated by free-form prompts. The first column contains images selected by *UniGen* as the test-time verifier.

Table 4: **Ablation of verification methods.**

| Method | Outcome | Rule | CoT | GenEval | DPG-Bench |
|--------|---------|------|-----|---------|-----------|
| *UniGen* | ✗ | ✗ | ✗ | 0.74 | 85.02 |
| | ✓ | ✗ | ✗ | 0.74 | 85.00 |
| | ✗ | ✓ | ✗ | 0.75 | - |
| | ✗ | ✗ | ✓ | 0.78 | 85.19 |

Table 5: **Ablation of *CoT-V* post-training.**

| Method | *CoT-V* Post-train | GENEVAL | DPG-BENCH |
|--------|--------------------|---------|-----------|
| Show-o | ✗ | 0.64 | 76.32 |
| | ✓ | 0.66 | 77.09 |
| *UniGen* | ✗ | 0.74 | 84.89 |
| | ✓ | 0.78 | 85.19 |

**Second, *CoT-V* post-training is essential for strong test-time verification.** As shown in Table 5, directly using *UniGen* without *CoT-V* post-train leads to notable performance drop, especially for GENEVAL. This comparison demonstrates that *CoT-V* post-train is pivotal for CoT verification.

**Third, *CoT-V* can effectively generalize to other models.** We finetune Show-o with DPO and *CoT-V* with our generated data to boost its generation performance. Results in Table 5 show that *CoT-V* is a general technique that can also enhance Show-o's generation performance.

### 4.3.3 Ablation of DPO

We ablate the contribution of each data source and demonstrate the effectiveness of our DPO data on other unified models.

**First, every prompt source contributes positively to generation performance.** Table 6 shows that adding only PARM DPO data results in remarkable improvements (row1 vs. row2), while further adding T2I-Comp mainly benefits DPG-BENCH (row2 vs. row3). *UniGen-DPO* with all of three prompts, introduces the best overall performance (row3 vs. row4).

Table 6: **Ablation study of DPO.** The results are from *UniGen-DPO* without test-time scaling.

| Method | PARM | T2I-Comp | SA1B | GenEval | DPG-bench |
|--------|------|----------|------|---------|-----------|
| *UniGen* | ✗ | ✗ | ✗ | 0.63 | 82.75 |
| | ✓ | ✗ | ✗ | 0.73 | 83.48 |
| | ✓ | ✓ | ✗ | 0.72 | 84.09 |
| | ✓ | ✓ | ✓ | 0.74 | 84.89 |
| Show-o | ✗ | ✗ | ✗ | 0.56 | 71.70 |
| | ✓ | ✓ | ✓ | 0.64 | 76.32 |

**Second, our DPO data also largely improves Show-o, showing that it is generalizable to other unified models.** When fine-tuning Show-o directly with our DPO data, we also observe a notable gain, from 0.56 to 0.64 on GENEVAL and from 71.70 to 76.32 on DPG-BENCH as shown in Table 6.

## 4.4 Ablation of SFT

Table 7: **Ablation of SFT stage.** PT-2 Data denotes the training data used in the PT-2 Stage. JD and TI denote JourneyDB and text-2-image-2M, respectively. The results are from *UniGen-SFT*.

| Und Data | Gen Data | GenEval | DPG-Bench | AI2D | GQA | POPE | MMMU | MathVista | RW-QA | ScienceQA | Seedbench | Und Avg. |
|---|---|---|---|---|---|---|---|---|---|---|---|---|
| SlowFast-LLaVA-1.5 | PT-2 Data | 0.56 | 79.67 | 68.3 | 62.4 | 87.5 | 33.3 | 42.2 | 54.4 | 79.6 | 70.7 | 62.3 |
|  | JD+TI | 0.63 | 82.77 | 68.0 | 62.5 | 87.4 | 32.4 | 45.2 | 58.6 | 79.7 | 71.1 | 63.1 |
| LLaVA1.5 | JD+TI | 0.64 | 81.82 | 48.7 | 62.8 | 87.4 | 27.1 | 22.1 | 53.7 | 55.5 | 64.0 | 52.7 |

By default, we use the image mixture from SlowFast-LLaVA-1.5 [90] as understanding datasets and JourneyDB and text-2-image-2M as the generation datasets. In this section, we ablate the datasets in Table 7 to evaluate their impacts and draw the following conclusion.

**First, using high-quality generation data is necessary for further lifting generation results.** JourneyDB and text-2-image-2M have much higher quality compared to the generation data used during the PT-2 stage. Table 7 (row1 vs. row2) shows that using high-quality generation data in the SFT stage results in better image generation performance.

**Second, using a stronger data mixture is crucial to improve the understanding performance, which is also helpful for fine-grained text-to-image generation.** As shown in Table 7 (row2 vs. row3), replacing SlowFast-LLaVA-1.5 mixture with LLaVA1.5's induces much worse understanding performance. Also, training with SlowFast-LLaVA-1.5 data produces higher results on DPG-BENCH. We believe a better understanding capability is important for comprehending the complex text prompts of DPG-BENCH that can eventually be beneficial for better text-to-image generation.

## 4.5 Ablation of PT-1 and PT-2

Table 8: **Impact of using understanding task in PT stages.** The results are from *UniGen-SFT*.

| Und Data PT-1 | Und Data PT-2 | GenEval | DPG-bench | AI2D | GQA | POPE | MMMU | MathVista | RW-QA | ScienceQA | Seedbench | Und Avg. |
|---|---|---|---|---|---|---|---|---|---|---|---|---|
| ✗ | ✗ | 0.61 | 82.51 | 60.5 | 59.6 | 87.4 | 30.9 | 38.1 | 49.0 | 72.0 | 66.1 | 58.0 |
| ✓ | ✓ | 0.63 | 82.75 | 68.0 | 62.5 | 87.4 | 32.4 | 45.2 | 58.6 | 79.7 | 71.1 | 63.1 |

Table 9: **Ablation of PT-1 stage.** Cls and Recap indicate class names and high-quality captions are used for generating images, respectively. The results are from *UniGen-SFT*.

| Stage I | Gen Data | GenEval | DPG-bench | AI2D | GQA | POPE | MMMU | MathVista | RW-QA | ScienceQA | Seedbench | Und Avg. |
|---|---|---|---|---|---|---|---|---|---|---|---|---|
| ✗ | – | 0.64 | 82.26 | 70.3 | 62.5 | 87.9 | 33.7 | 45.7 | 54.2 | 80.5 | 71.7 | 63.3 |
| ✓ | ImageNet(Cls) | 0.63 | 82.75 | 67.0 | 62.5 | 88.0 | 31.4 | 41.4 | 53.6 | 79.8 | 71.1 | 61.8 |
| ✓ | ImageNet(Recap) | 0.63 | 82.75 | 68.0 | 62.5 | 87.4 | 32.4 | 45.2 | 58.6 | 79.7 | 71.1 | 63.1 |

We explore the necessity and key factors of *UniGen*'s pre-training stages. We first discuss whether we should include the understanding dataset in the pre-training stages as shown in Table 8. Second, we ablate the impact of the generation datasets to both generation and understanding performance in Table 9 (for PT-1) and Table 10 (for PT-2). Since PT-1 and PT-2 are early stages in *UniGen*'s training pipeline, we continue the training to the SFT stage to verify their impact on the final performance more reliably. Unless noted otherwise, all ablations in this section use *UniGen*'s default SFT settings. Here we highlight the following observations.

**First, including understanding data in pre-training stages is crucial for both generation and understanding performance.** In Table 8's row 1, we keep the default setting and only remove the understanding loss from the training objectives. We observe a significant performance decrease across generation and understanding benchmarks at the SFT stage. We attribute this to the fact that understanding data is important for a better vision-language alignment in early training stages, which is helpful for both image-to-text and text-to-image tasks.

**Second, the high-quality text-to-image task is more effective than the de facto class-to-image task in PT-1.** One common practice of unified MLLMs for pre-training is using the class-to-image task with ImageNet [84, 87]. However, we find that using ImageNet with fine-grained captions leads to better performance for understanding tasks in the *UniGen-SFT* stage as shown in Table 9 (row2 vs. row3). This is a result of better vision-language alignment introduced by the detailed caption-to-image mapping.

Table 10: **Ablation of PT-2 stage.** The reported results are from *UniGen-SFT*.

| Stage II | Gen & Und Data | GenEval | DPG-bench | AI2D | GQA | POPE | MMMU | MathVista | RW-QA | ScienceQA | Seedbench | Und Avg. |
|---|---|---|---|---|---|---|---|---|---|---|---|---|
| ✗ | – | 0.58 | 79.25 | 70.2 | 61.9 | 87.3 | 31.6 | 47.0 | 54.9 | 82.5 | 71.2 | 63.3 |
| ✓ | CC+SA+IMN | 0.59 | 80.64 | 64.5 | 61.6 | 87.9 | 30.8 | 40.9 | 54.2 | 77.0 | 69.6 | 60.8 |
| ✓ | (SA+IMN)(Recap) | 0.64 | 82.63 | 67.2 | 62.4 | 87.5 | 31.0 | 40.8 | 56.5 | 79.7 | 71.1 | 62.0 |
| ✓ | (CC+IMN)(Recap) | 0.63 | 82.75 | 67.9 | 62.1 | 87.8 | 31.6 | 42.0 | 58.2 | 80.3 | 70.8 | 62.6 |
| ✓ | (CC+SA)(Recap) | 0.62 | 82.34 | 68.6 | 62.2 | 87.4 | 30.6 | 45.4 | 59.3 | 80.5 | 70.8 | 63.1 |
| ✓ | (CC+SA+IMN)(Recap) | 0.63 | 82.75 | 68.0 | 62.5 | 87.4 | 32.4 | 45.2 | 58.6 | 79.7 | 71.1 | 63.1 |

**Third, to maintain high performance on generation, we need both PT-1 and PT-2.** According to Table 9 (row1 vs. row3) and Table 10 (row1 vs. row6), we notice that completely removing PT-1 or PT-2 stage will largely decrease the generation metrics. Especially, eliminating PT-2 has a much bigger negative impact, leading to a dramatic drop of numbers on both GENEVAL and DPG-BENCH. Excluding PT-1 will not apparently affect GENEVAL, but hurts DPG-BENCH. This is because the prompts of DPG-BENCH are more complicated, thus more pre-training helps our model to better comprehend their semantics.

**Fourth, to keep a strong understanding performance, we need at least one of the PT-1 and PT-2.** According to Table 8, we infer that the understanding performance will be destroyed if we remove both PT-1 and PT-2. However, discarding PT-1 or PT-2 in Table 9 and Table 10 does not impact understanding numbers. As a result, we recommend keeping at least one of them for good understanding capability and leveraging both of them for the best generation and understanding performance if the compute budget allows.

**Fifth, using high-quality captions in PT-2 is important for understanding and generation performance.** Table 10 (row2 vs. row6) demonstrates that using high-quality image captions results in stronger performance in both understanding and generation tasks. This is due to the better text-to-image and image-to-text alignment learned from the fine-grained captions.

**Sixth, each data source of PT-2 has meaningful contributions.** We remove each data component from the training set of PT-2 and observe that retaining all of them leads to the best performance as shown in Table 10 (row3 to row6). This finding supports the usefulness of each dataset we curated.

## 5 Conclusion

We present *UniGen*, an MLLM for unified multimodal understanding and generation. We discuss the key factors along the entire training pipeline and propose optimization methods to improve the performance. We also make the first attempt to collaborate *UniGen*'s understanding and generation capabilities, by enabling *UniGen* to perform as both image generator and verifier during test-time scaling. As a result, we successfully further boost the image generation quality by a clear margin. Trained with only open-source datasets, *UniGen* achieves the state-of-the-art performance across extensive understanding and generation benchmarks. We hope our exploration and ablation studies provide insights into the future development of strong unified MLLMs.

**Limitation.** *First*, we instantiate *UniGen* with only a 1.5B model, since larger scales will impose much higher demands on the computational cost. However, larger models have been shown effective for improving both understanding and generation performance [9]. *Second*, our generation capability targets at promoting semantic alignment between the input text prompt and the generated image, therefore we only focus on a resolution of $256 \times 256$. We plan to support higher resolution image generation, such as 480p or even 1080p, which is valuable for improving the visual fidelity. *Third*, although achieving convincing results on DPG-Bench, *CoT-V* is still limited for complicated text prompts, due to the noisy CoT data generated by Qwen2.5VL as a pseudo labeler. This could be largely relieved by using a stronger pseudo labeler or leveraging human filtering in the future. Equipping *UniGen* with stronger reasoning and CoT capabilities in an earlier stage is also a promising direction.

**Broader Impact.** Unified MLLMs offer scientific benefits by enabling human-AI interaction and advancing general-purpose multimodal understanding. There are many real-world applications, such as design assistants, education, and collaborative robots. However, there could be unintended usages and we advocate responsible usage complying with applicable laws and regulations.

**Acknowledgment.** We thank Haiming Gang, Jesse Allardice, Shiyu Li, and Yifan Jiang for their kind help.

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

# A  Benchmarks and Evaluation Protocol

**For image understanding**, we include widely-used *(i)* general VQA benchmarks, such as GQA [26], RealWorld-QA [18], and Seedbench [33], *(ii)* knowledge-based benchmarks, such as AI2D [30], MMMU [99], and MathVista [46], and *(iii)* hallucination benchmarks, such as POPE [36]. We leverage the `lmms-eval`[1] toolkit to compute the results for the above benchmarks.

**For text-to-image generation benchmarks**, we report results on GENEVAL [17] and DPG-BENCH [23] to comprehensively evaluate the semantic alignment between a text prompt and the generated images. To fairly compare with recent unified MLLMs [84, 9, 87], our results are obtained using the official evaluation repository of GENEVAL[2] and DPG-BENCH[3].

# B  More Ablation Studies

Table 11: **Ablation of Best-of-N strategy.** The results are from *UniGen* with *CoT-V*.

| $N$ | GenEval | DPG-Bench | Speed (s/img) |
|---|---|---|---|
| 1 | 0.73 | 84.89 | 3.55 |
| 3 | 0.77 | 85.13 | 16.59 |
| 5 | 0.78 | 85.19 | 27.22 |

Table 12: **Ablation of the order of *CoT-V* post-training and DPO training.** The results are from *UniGen* with *CoT-V*.

| Sequence | Und Avg. | GenEval | DPG-bench |
|---|---|---|---|
| DPO $\rightarrow$ *CoT-V* Post-train | 62.7 | 0.78 | 85.19 |
| *CoT-V* Post-train $\rightarrow$ DPO | 62.7 | 0.76 | 85.20 |

**Choosing $N$ for Best-of-N strategy.** We ablate $N$ for Best-of-$N$ selection with *CoT-V*. As shown in Table 11, using larger $N$ consistently yields higher performance on GENEVAL benchmark. To assess the efficiency trade-off, we further measure the inference speed of *CoT-V* on the same H100 GPU, considering both MLLM execution time and tokenizer decoding overhead. The average speed is computed across all GENEVAL samples using one prompt per batch. Under the default setting of (N=5), the total inference time is approximately $8\times$ slower than standard inference, highlighting the trade-off between increased computational cost and improved accuracy.

**Switching the order of CoT-V post-training and DPO training.** We reverse the order of DPO training and CoT-V post-training and present the results in Table 12. The findings indicate that the training order has minimal impact, with the default setting showing a slight advantage on GENEVAL.

Table 13: **Ablation of visual tower for understanding.** The results are from *UniGen*-SFT.

| Visual Encoder | Und Avg. | GenEval | DPG-Bench |
|---|---|---|---|
| Freeze | 63.11 | 0.63 | 82.75 |
| Unfreeze | 63.16 | 0.65 | 82.71 |

Table 14: **Ablation of visual tokenizer for generation.** The results are from *UniGen*-SFT.

| Tokenizer | GenEval | DPG-bench |
|---|---|---|
| MAGViTv2 | 0.63 | 82.75 |
| VQ-16 | 0.62 | 82.93 |

**Freezing the visual tower for understanding** during supervised fine-tuning achieves performance on par with the unfrozen setting on understanding benchmarks. As reported in Table 13, unfreezing the visual tower leads to a 2% improvement on GENEVAL, while freezing the encoder reduces computational cost. These results indicate that the frozen design leads to a more cost-efficient option with only minor performance differences.

**Changing the discrete visual tokenizers.** By default, we adopt the MAGViTv2 implementation from Show-o[4]. To ablate the impact of discrete visual tokenizer on generation, we further experiment with the VQ-16 tokenizer from LLamaGen [65]. As shown in Table 14, both tokenizers achieve comparable performance on GENEVAL and DPG-BENCH after supervised fine-tuning, demonstrating the robustness and generalizability of our training framework.

---

[1]https://github.com/EvolvingLMMs-Lab/lmms-eval
[2]https://github.com/djghosh13/geneval/tree/main
[3]https://github.com/TencentQQGYLab/ELLA/tree/main
[4]https://huggingface.co/showlab/magvitv2

## C  More Results

We present the breakdown comparison of *UniGen* against state-of-the-art models on GENEVAL and DPG-BENCH in Table 15 and Table 16.

Table 15: **Comparison with state-of-the-art models on the GenEval benchmark.**

| Model | #Params | Single Obj. | Two Obj. | Counting | Colors | Position | Color Attri. | Overall↑ |
|---|---|---|---|---|---|---|---|---|
| *Text-to-Image Generation Models* | | | | | | | | |
| DALLE-2 [61] | 6.5B | 0.94 | 0.66 | 0.49 | 0.77 | 0.10 | 0.19 | 0.52 |
| DALLE-3[4] | - | 0.96 | 0.87 | 0.47 | 0.83 | 0.43 | 0.45 | 0.67 |
| Emu3 [80] | 8B | 0.98 | 0.71 | 0.34 | 0.81 | 0.17 | 0.21 | 0.54 |
| SDXL [57] | 2.6B | 0.98 | 0.74 | 0.39 | 0.85 | 0.15 | 0.23 | 0.55 |
| SimpleAR [76] | 1.5B | - | 0.90 | - | - | 0.28 | 0.45 | 0.63 |
| Infinity [21] | 2B | - | 0.85 | - | - | 0.49 | 0.57 | 0.73 |
| *Unified MLLMs* | | | | | | | | |
| Show-o [87] | 1.3B | 0.95 | 0.52 | 0.49 | 0.82 | 0.11 | 0.28 | 0.53 |
| Janus [84] | 1.3B | 0.97 | 0.68 | 0.30 | 0.84 | 0.46 | 0.42 | 0.61 |
| Janus-Pro [9] | 1.5B | 0.98 | 0.82 | 0.51 | 0.89 | 0.65 | 0.56 | 0.73 |
| ILLUME [74] | 7B | 0.99 | 0.86 | 0.45 | 0.71 | 0.39 | 0.28 | 0.61 |
| UniToken [29] | 7B | 0.99 | 0.80 | 0.35 | 0.84 | 0.38 | 0.39 | 0.63 |
| VARGPT-v1.1 [107] | 9B | 0.96 | 0.53 | 0.48 | 0.83 | 0.13 | 0.21 | 0.53 |
| TokenFlow-XL [58] | 13B | 0.93 | 0.72 | 0.45 | 0.82 | 0.45 | 0.42 | 0.63 |
| *UniGen* | 1.5B | 1.00 | 0.92 | 0.68 | 0.87 | 0.48 | 0.52 | **0.74** |
| *UniGen + CoT-V* | 1.5B | 1.00 | 0.94 | 0.78 | 0.87 | 0.57 | 0.54 | **0.78** |

Table 16: **Comparison with state-of-the-art models on the DPG-bench benchmark**.

| Model | #Params | Global | Entity | Attribute | Relation | Other | Overall↑ |
|---|---|---|---|---|---|---|---|
| *Text-to-Image Generation Models* | | | | | | | |
| Hunyuan-DiT [37] | - | 84.59 | 80.59 | 88.01 | 74.36 | 86.41 | 78.87 |
| DALLE-3[4] | - | 90.97 | 89.61 | 88.39 | 90.58 | 89.83 | 83.50 |
| Emu3 [80] | 8B | 85.21 | 86.68 | 86.84 | 90.22 | 83.15 | 80.60 |
| SDXL [57] | 2.6B | 83.27 | 82.43 | 80.91 | 86.76 | 80.41 | 74.65 |
| SimpleAR [76] | 1.5B | 87.97 | - | - | 88.33 | - | 81.97 |
| Infinity [21] | 2B | 93.11 | - | - | 90.76 | - | 83.46 |
| *Unified MLLMs* | | | | | | | |
| Show-o* [87] | 1.3B | 80.39 | 80.94 | 82.17 | 83.36 | 82.88 | 71.70 |
| Janus [84] | 1.3B | 82.33 | 87.38 | 87.70 | 85.46 | 86.41 | 79.68 |
| Janus-Pro [9] | 1.5B | 87.58 | 88.63 | 88.17 | 88.98 | 88.30 | 82.63 |
| VARGPT-v1.1 [107] | 9B | 84.83 | 82.80 | 84.95 | 88.13 | 87.70 | 78.59 |
| TokenFlow-XL [58] | 13B | 78.72 | 79.22 | 81.29 | 85.22 | 71.20 | 73.38 |
| *UniGen* | 1.5B | 91.53 | 90.39 | 90.30 | 91.09 | 90.86 | **84.89** |
| *UniGen + CoT-V* | 1.5B | 91.95 | 89.68 | 90.90 | 92.04 | 90.91 | **85.19** |

## D  Details of Test-Time Strategies

### D.0.1  Prompts of different verifications for test-time inference

> **Prompt. 1: Chain-of-Thought Verification**
>
> {image} This image is generated by a prompt: {prompt}. Please assess the image generation quality step by step. First, breakdown the prompt into multiple visual questions and iteratively answer each question with Yes or No between <think_start> <think_end>. Questions should cover all-round details about whether the image accurately represents entity categories, counting of entities, color, spatial relationship in the prompt. Next, output the final result between <answer_start> <answer_end>. Output Yes if all multi-choice answers equal yes to show the image has accurate alignment with the prompt. Otherwise answer with No.

> **Prompt. 2: Outcome Verification**
>
> {image} This image is generated by a prompt: {prompt}. Does this image accurately represent the prompt? Please answer yes or no.

> **Prompt. 3: Rule-based Verification**
>
> {image} {question} Please answer yes or no with detail explanation.

# E  Details of Training

Table 17: **Hyperparameter setup for different training stages of *UniGen*.** Data ratio refers to the ratio of image understanding data, pure text data, and image generation data.

| Hyperparameters | PT-1 | PT-2 | SFT | DPO | *CoT-V* Post-Training |
|---|---|---|---|---|---|
| Learning rate | $1.0 \times 10^{-4}$ | $1.0 \times 10^{-4}$ | $1.0 \times 10^{-3}$ | $1.0 \times 10^{-5}$ | $1.0 \times 10^{-5}$ |
| LR scheduler | Cosine | Cosine | Cosine | Cosine | Cosine |
| Weight decay | 0.01 | 0.01 | 0.05 | 0.05 | 0.05 |
| Gradient clip | 1.0 | 1.0 | 1.0 | 1.0 | 1.0 |
| Optimizer | AdamW | AdamW | AdamW | AdamW | AdamW |
| Warm-up steps | 6000 | 5000 | 1000 | 500 | 0 |
| Training steps | 150k | 400k | 146k | 1.6k | 0.5k |
| H100 hours | 1.0k | 2.8k | 240 | 5 | 0.7 |
| Batch size | 896 | 512 | 64 | 80 | 64 |
| Data ratio | 2:1:4 | 2:1:4 | 4:1:3 | -:-:1 | 1:-:- |

### E.0.1  Training Parameters

Details of hyperparameters during each training stage are presented in Table 17.

### E.0.2  Training Data Overview

We list the datasets used in our training stages in Table 18. Refer to Appendix E.0.4 and Appendix E.0.5 for more details about the preference data and *CoT-V* data.

### E.0.3  Prompts for Generating Pre-Train Data

We use Prompt. 4 to prompt Qwen2.5VL-7B for generating fine-grained captions for CC-3M, CC-12M, SA-1B and ImageNet that are used in pre-training stages as shown in Table 18.

> **Prompt. 4: Re-caption**
>
> {image} What is the content of this image?

### E.0.4  Preference Data Generation for DPO

**PARM.** GENEVAL metric is used to rate each generated image candidate per prompt. The highest and lowest rated ones are used as the preferred and rejected samples for this prompt.

**T2I-Comp and SA-1B.** These prompts are more complex than the prompts of PARM, therefore, it is difficult to rate each generated image using rule-based metrics. We adopt a two-step approach to evaluate the coherence between an image and the prompt. First, we use Qwen2.5-7B to decompose each text prompt into atomic facts represented as questions using Prompt. 5. Then, the image with each decomposed question is fed into Qwen2.5VL-7B with Prompt. 6. The model responds with *yes* if visual generation passes the fact-check and with *no* otherwise. The final score of an image is calculated by averaging the results of all fact-checks. We take the most and least aligned images per prompt as the preferred and rejected sample pairs.

> **Prompt. 5: Visual Questions Generation**
>
> Now you need to convert an image description into fine-grained, related visual questions. The questions should comprehensively cover detailed visual facts of entities, attributes (e.g., color, count, texture, shape, and size), and relationships (e.g., spatial and non-spatial) between the entities mentioned in the description. Please complete the task by analyzing each clause in the sentence step by step. For each clause, first raise questions about whether each mentioned entity exists in the image. Then, raise questions about whether the attributes or relationships of the entities are accurately represented in the image. For an image accurately aligned with the description, all questions should be answered with "yes"; otherwise, they should be answered with "no".
> Make sure all questions are able to be responded with yes or no and are connected with semicolon. Here are examples:
> Example 1:
>    *description*: three black keys, four chickens and a fabric blanket
>    *output*: Are there keys?; Are there three keys?; Are the keys black?; Are there chickens?; Are there four chickens?; Is there a blanket?; Is the blanket fabric?
> Example 2:
>    *description*: A person in a blue shirt and red and black apron is using a power tool, likely a drill, to assemble a white cabinet or shelving unit indoors. The floor is covered with light-colored wood or laminate material.
>    *output*: Is there a person?; Is the person wearing a shirt; Is the shirt blue?; Is the person wearing a apron?; Is the apron red and black?; Is the person using a drill?; Is there a white cabinet or shelving unit?; Is the person using the drill indoors?; Is there light-colored wood on the floor?; Is there laminate material on the floor?
> Example 3:
>    *description*: a large Ferris wheel with a digital clock showing the time as 11:00. The Ferris wheel is located in an urban area, as indicated by the modern buildings in the background. There is also a tree on the left side of the image, partially obscuring the view of the Ferris wheel. The sky appears clear, suggesting a sunny day.
>    *output*: Is there a Ferris wheel?; Is there a digital clock?; Is the digital clock on the Ferris wheel?; Is the digital clock showing the time as 11:00?; Is the Ferris wheel located in an urban area?; Are there modern buildings in the background?; Is there a tree on the left side?; Is the sky clear and sunny?
> Please convert this image description:{description}into fine-grained related visual questions.

> **Prompt. 6: Visual Fact-Check**
>
> {image} {question} Please answer yes or no without explanation.

We display some visual results of DPO preference data in Fig. 6.

Table 18: **Training data overview.** CC, SA, IMN, JD, T2I indicate CC-3M&CC-12M, SA-1B, ImageNet, JourneyDB, text-2-image-2M, respectively. Recap denotes that the images are re-captioned using Qwen2.5VL-7B.

| Stage | Gen Data | Und Data | Text-only |
|---|---|---|---|
| PT-1 | IMN (Recap) | (CC+SA+IMN) (Recap) | RefinedWeb |
| PT-2 | (CC+SA+IMN) (Recap) | (CC+SA+IMN) (Recap) | RefinedWeb |
| SFT | JD+T2I | SF-LLaVA1.5 (Image Mixture) [90] | RefinedWeb |
| DPO | Preference Data | – | – |
| *CoT-V* | – | *CoT-V* data | – |

### E.0.5 *CoT-V* Post-Training Data

We sample 20K preference data from PARM and T2I-Comp in Appendix E.0.4 to construct our *CoT-V* post-train data and use Prompt. 1 to encourage *UniGen* to generate CoT reasoning during training. To supervise the training process, we construct the CoT reasoning labels based on decomposed

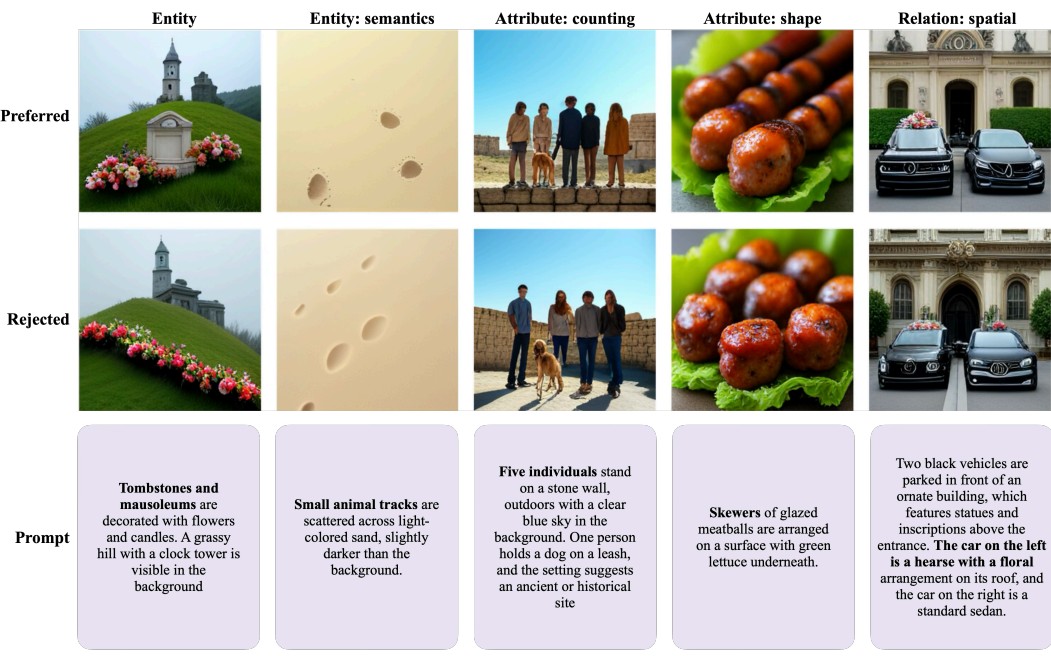

Figure 6: **Visual examples of our generated preference data for DPO training.**

atomic question-answer pairs corresponding to visual facts presented in the image. For PARM, we separate each prompt into fine-grained sub-questions according to the templates originally used for generating the prompt. Rules of GENEVAL are used to label each sub-question corresponding to the image with *yes* or *no*. For T2I-Comp, we directly use the decomposed question-answers from the preference data. The final answer is *yes* if all the sub-questions are answered with *yes* and it is *no* otherwise. To form the CoT label, the separated question-answers are treated as a thinking process enclosed within special tokens `<think_start><think_end>`, and the final answer resides within `<answer_start><answer_end>`.

## F   More Qualitative Results

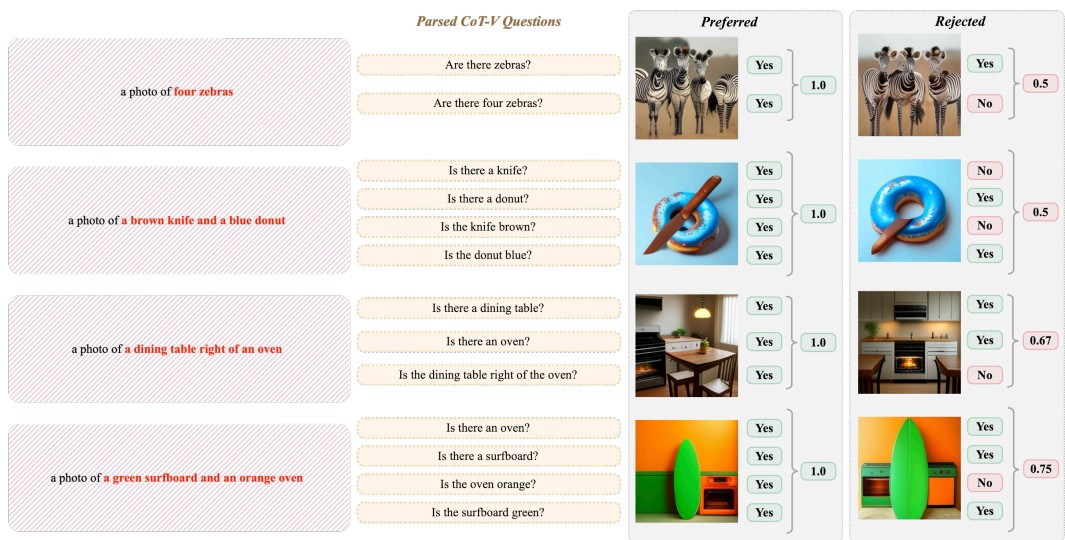

Figure 7: **Successful examples and *CoT-V* verification on GENEVAL.**

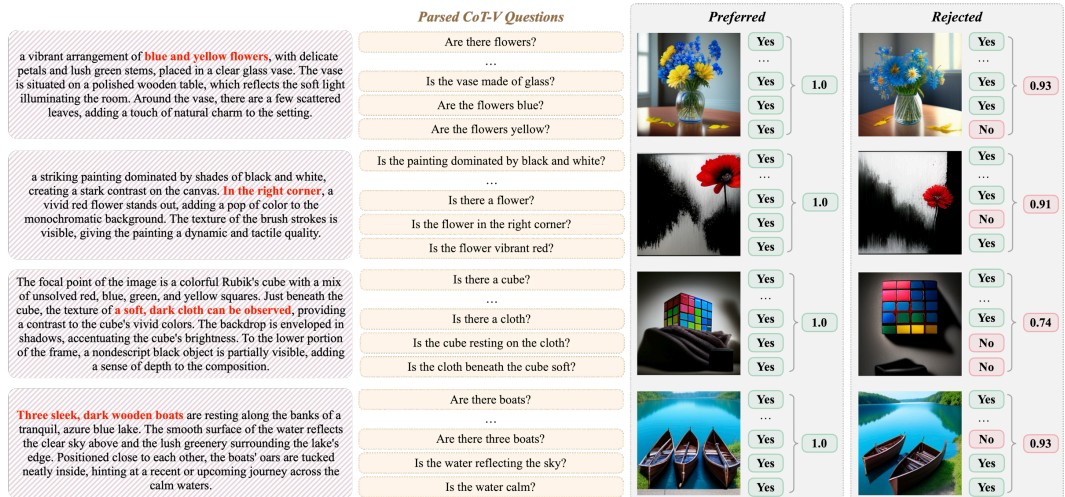

Figure 8: **Successful examples and *CoT-V* verification on DPG-BENCH.**

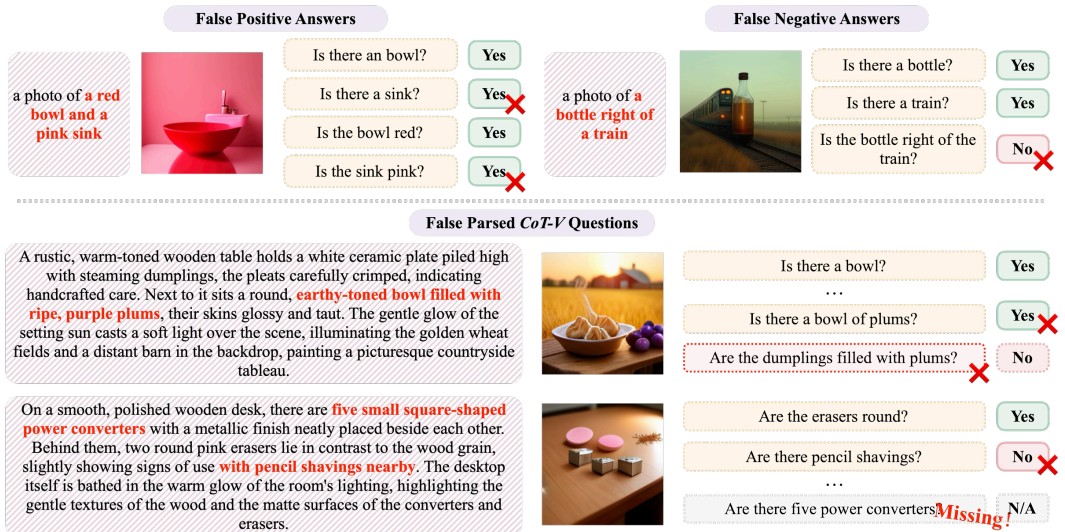

Figure 9: **Failure cases on GENEVAL and DPG-BENCH.** The **top half** of the image shows failed examples of *CoT-V* on short prompts. The **bottom half** of the image shows additional cases with bad or missing questions when *CoT-V* parses the complicated and long prompts.

We present qualitative results in Fig. 7 and Fig. 8. They indicate *CoT-V*'s effectiveness on selecting images that accurately convey the entities, color, counting and spatial relation. However, as failure cases shown in Fig. 9, *CoT-V* may struggle with hallucination in more difficult cases. Particularly, we acknowledge that *UniGen* still falls short of generating an accurate reasoning process given free-form complex prompts. We posit that scaling up the model size or improving our CoT training via reinforcement learning algorithms could improve the capability of reasoning and image generation, and consequently enhance the overall performance of *CoT-V*.

