# OpenReview forum: "UniGen: Enhanced Training & Test-Time Strategies for Unified Multimodal Understanding and Generation"
_NeurIPS.cc/2025/Conference — NeurIPS 2025 poster_

### Official Review · Reviewer_CnR8 · 2025-06-22

**Clarity:** 3
**Significance:** 3
**Originality:** 2
**Rating:** 4
**Confidence:** 4

**Summary:**

The paper presents UniGen, a unified multimodal large language model (MLLM) for image understanding and generation. It proposes a comprehensive training pipeline including pretraining, supervised fine-tuning, and direct preference optimization. A test-time Chain-of-Thought Verification (CoT-V) method is introduced to enhance generation quality through self-verification and Best-of-N selection.

**Questions:**

1. **Comparison Between Model Scaling and Test-Time Scaling:** I’m curious about the trade-offs between scaling the model size (e.g., training a larger MLLM) versus applying test-time strategies like CoT-V with a smaller model. How do they compare in terms of performance gains and computational overhead during inference?

2. **Open-Source Concern:** Since a major contribution of this work is the unified training pipeline and the use of open-source data (with additional synthetic annotations), there is a concern about whether the authors will release the training code and datasets. Making these resources publicly available would greatly benefit reproducibility and further research.

**Ethical Concerns:**

["NO or VERY MINOR ethics concerns only"]

**Limitations:**

Yes

**Quality:**

3

**Strengths And Weaknesses:**

Strengths:

1. **Clear Pipeline:** The paper outlines a clear and systematic methodology, covering multi-stage pretraining, supervised fine-tuning, preference optimization, and test-time reasoning. This end-to-end design provides practical insights into constructing unified multimodal large language models.

2.  **Good Empirical Results:** UniGen achieves notable improvements over prior unified MLLMs on multiple benchmarks, including GENEVAL and DPG-BENCH, demonstrating the effectiveness of its design and training strategies.

Weaknesses:

1. **Limited Scale:**  The current model is constrained to 1.5 billion parameters. While the results are competitive, extending the framework to larger model sizes could further strengthen the contribution.

2. **High Test-Time Overhead:** The proposed Chain-of-Thought Verification (CoT-V) strategy involves evaluating multiple candidates per prompt, which may be computationally expensive. A detailed analysis of test-time cost and latency would improve the practicality assessment of the method.

---

> ### Author Rebuttal · Authors · 2025-07-31
>
> **To Reviewer CnR8**
>
> Thanks for your helpful comments. Please see our responses below.
>
> **Q1:** Limited Scale: The current model is constrained to 1.5 billion parameters. While the results are competitive, extending the framework to larger model sizes could further strengthen the contribution.
>
> **A1:** To resolve your concern, we scale up our model to 7B parameters. As shown in the table, we observe improvement on both understanding and generation benchmarks with the larger model size. Note that due to the limited time, we use the DPO and CoT-V data generated by UniGen-1.5B for training UniGen-7B. We expect more increase if we use UniGen-7B for creating its own data.
>
> | Model          | Und Avg.|GenEval | DPG-Bench |
> | -------------- | ------- |------- | --------- |
> | UniGen-1.5B | 62.66| 0.78    | 85.19     |
> | UniGen-7B | 63.78| 0.79    | 85.79     |
>
> **Q2:** High Test-Time Overhead: The proposed Chain-of-Thought Verification (CoT-V) strategy involves evaluating multiple candidates per prompt, which may be computationally expensive. A detailed analysis of test-time cost and latency would improve the practicality assessment of the method.
>
> **A2:** We test the speed of image generation with different inference strategies on the same H100 GPU, covering both the MLLM running time and the tokenizer decoding cost.
> We compute the averaged speed on all samples of GenEval by feeding our model with one prompt in each batch.
> For CoT-V, we output N candidate images for each prompt, and perform self-verification by sequentially evaluating each image-prompt pair.
> Then, we select the best image out of N candidates. By default (N=5), the total inference time for picking the best image cost around 27.22s. As shown in the table, we observe improved performance as we using larger N. This aligns with the goal of test-time scaling that is trading compute for accuracy.
>
> | Model          |   N for BoN    | GenEval | DPG-Bench | Speed (s/img) |
> | -------------- | ------- | ------- | --------- | ---- |
> | UniGen-1.5B |   1   |  0.73   |   84.89    |  3.55   |
> | UniGen-1.5B |   3   |  0.77   |       85.13     |  16.59  |
> | UniGen-1.5B |   5   |  0.78   |   85.19    |  27.22  |
>
> **Q3:** Comparison Between Model Scaling and Test-Time Scaling: I’m curious about the trade-offs between scaling the model size (e.g., training a larger MLLM) versus applying test-time strategies like CoT-V with a smaller model. How do they compare in terms of performance gains and computational overhead during inference?
>
> **A3:** We add the experiment of 7B model. As shown in the table below, with N set to 3, CoT-V effectively bridges the gap between 7B and 1.5B model with slightly slower inference speed than the 7B model.
>
> | Model          |   N for BoN    | GenEval | DPG-Bench | Speed (s/img) |
> | -------------- | ------- | ------- | --------- | ---- |
> | UniGen-1.5B |   1   |  0.73   |   84.89    |  3.55   |
> | UniGen-1.5B |   3   |  0.77   |       85.13     |  16.59  |
> | UniGen-1.5B |   5   |  0.78   |   85.19    |  27.22  |
> | UniGen-7B   |   1   |  0.75   |   85.71    |  12.92  |
>
> **Q4:** Open-Source Concern: Since a major contribution of this work is the unified training pipeline and the use of open-source data (with additional synthetic annotations), there is a concern about whether the authors will release the training code and datasets. Making these resources publicly available would greatly benefit reproducibility and further research.
>
> **A4:** We will open source the code, the curated dataset and the script for genearting synthetic annotations.

---

> > ### Comment · Reviewer_CnR8 · 2025-08-07
> >
> > Thank you for the detailed response. My concerns have been addressed, and I’ll keep my score.

---

> > > ### Author Response · Authors · 2025-08-07
> > >
> > > Thank you!

---

### Official Review · Reviewer_WbDR · 2025-06-22

**Clarity:** 3
**Significance:** 3
**Originality:** 4
**Rating:** 4
**Confidence:** 4

**Summary:**

This paper introduces a unified image understanding and generation model UniGen. This model borrows the ideas from some established models (e.g., Janus, Show-o) to constitute its architecture and undergoes five training stages: pre-training, supervised fine-tuning (SFT), direct preference optimization, and test-time scaling learning. The authors propose a post-training approach named Chain-of-Thought Verification, which uses UniGen itself as the verifier to evaluate the generated images and produce CoT data. They then conduct SFT on the CoT data to endow the model with thinking ability. In experiments, UniGen surpasses other MLLMs by notable margins.

**Questions:**

See weaknesses.

**Ethical Concerns:**

["NO or VERY MINOR ethics concerns only"]

**Final Justification:**

Most of the concerns are addressed. Therefore, the rating will be maintained.

**Limitations:**

Yes.

**Paper Formatting Concerns:**

No.

**Quality:**

3

**Strengths And Weaknesses:**

> Strengths:

1. Unifying image understanding and generation with one MLLM has been a hot and interesting topic in recent years. This paper provides a new unified MLLM with SOTA performance for the community, which can advance this field.
2.  The proposed CoT-V method is novel and effective. The idea of asking the model to answer a series of atomic questions to serve as the thinking process is interesting and can intuitively enhance model generation performance.
3. The effectiveness of CoT-V is not only grounded in UniGen but also in unified models, indicating this technique’s generalizability.
4. The constructed preference dataset would be a valuable contribution to the community if published.

> Weaknesses:

1. The resolution of Fig. 1 is low. Please make a refinement.
2. In Introduction, the authors state that “we walk through the entire life cycle of the model development” and “draw insightful lessons”. However, in experiments, the authors seem not to show what these “lessons” are. They primarily focus on demonstrating the effectiveness of their specific model design or training strategies, rather than introducing some globally useful knowledge or experiences that can benefit the community in designing such sophisticated models. Can the authors share some experiences (the learned “lessons”) regarding training or model design from a more global perspective?
3. It is unclear whether the model weights and curated dataset in this paper will be published. If so, the authors should make a clear statement somewhere in the main text.
4. In DPO training, what is the size of preferred samples $y_w$ and rejected samples $y_l$ used in experiments?
5. In L193, the authors say that “RV struggles with free-form or complex instructions, such as those in DPG-Bench [21].” and choose CoT-V as a better option. The description is too limited to demonstrate RV’s drawbacks. Since the Best-of-N strategy can also be combined with RV and achieve similar functionality, the superiority of CoT-V over RV remains unclear. Can the authors give some examples of free-form or complex instructions to better illustrate RV’s limitation and give more details regarding CoT-V’s superiority?
6. The authors use UniGen-DPO as a self-verifier to produce training data for test-time scaling. The model can generate low-quality or erroneous data, and it seems that no human correction has been applied to this data. How do the authors ensure data quality, and will low-quality data hamper model learning?
7. During training, UniGen first undergoes DPO and then the test-time scaling training. However, it seems that the model could first undergo the test-time scaling stage and then DPO, since training with CoT-V is simply supervised fine-tuning. This also makes sense to first foster the model’s CoT-based reasoning ability and then optimize its preference. What is the reason behind choosing the original order? Will this lead to better model performance? I recommend the authors provide more clarification on this point.
8. typo: L257: stages -> stages.

---

> ### Author Rebuttal · Authors · 2025-07-31
>
> **To Reviewer WbDR**
>
> Thanks for your suggestions. Please see our responses below.
>
> **Q1:** The resolution of Fig. 1 is low. Please make a refinement.
>
> **A1:** Thanks for the suggestion. We will adjust it in the final draft.
>
> **Q2:** In Introduction, the authors state that “we walk through the entire life cycle of the model development” and “draw insightful lessons”. However, in experiments, the authors seem not to show what these “lessons” are. They primarily focus on demonstrating the effectiveness of their specific model design or training strategies, rather than introducing some globally useful knowledge or experiences that can benefit the community in designing such sophisticated models. Can the authors share some experiences (the learned “lessons”) regarding training or model design from a more global perspective?
>
> **A2:** Thanks for pointing this out. We have provided valuable lessons in the supplementary materials (section B). We highlighted the lessons we want to share in bold font. I summarize them in the following.
>
> L1: using high-quality generation data is necessary for further lifting generation results (line 932-935).
>
> L2: using a stronger data mixture is crucial to improve the understanding performance, which is also helpful for fine-grained text-to-image generation (line 936-941).
>
> L3: including understanding data in pre-training stages is crucial for both generation and understanding performance (line 950-955).
>
> L4: the high-quality text-to-image task is more effective than the de facto class-to-image task in PT-1 (line 956-961).
>
> L5: to maintain high performance on generation, we need both PT-1 and PT-2 (line 962-968).
>
> L6: to keep a strong understanding performance, we need at least one of the PT-1 and PT-2 (line 969-974).
>
> L7: using high-quality captions in PT-2 is important for understanding and generation performance (line 975-978).
>
> L8: each our selected data source of PT-2 has meaningful contributions (line 979-981). Future research is encouraged to start from this mixture.
>
>
> **Q3:** It is unclear whether the model weights and curated dataset in this paper will be published. If so, the authors should make a clear statement somewhere in the main text.
>
> **A3:** We will open source the code and the curated datasets, and will add the statement to the main text.
>
> **Q4:** In DPO training, what is the size of preferred samples and rejected samples used in experiments?
>
> **A4:** As stated in line 150/154/159, we collected 8k short, 8k medium and 8k long prompts, respectively. After filtering, we employ around 13k triplets of prompts, perferred images and rejected images. We wil add more details in the final draft.
>
> **Q5:** In L193, the authors say that “RV struggles with free-form or complex instructions, such as those in DPG-Bench [21].” and choose CoT-V as a better option. The description is too limited to demonstrate RV’s drawbacks. Since the Best-of-N strategy can also be combined with RV and achieve similar functionality, the superiority of CoT-V over RV remains unclear. Can the authors give some examples of free-form or complex instructions to better illustrate RV’s limitation and give more details regarding CoT-V’s superiority?
>
> **A5:** For almost all the examples in DPGBench, the prompts are free-form and complicated (see a typical example below). It is impossible to design a general rule that can decompose the prompts into atomic questions.
>
> ```
> Three sleek remotes are aligned perfectly next to a simple black picture frame, which encloses a monochrome photograph. These objects rest on the glossy surface of a rich mahogany coffee table. Surrounding them, the soft glow of a nearby lamp illuminates the living room's plush, earth-toned furniture, casting subtle shadows that contribute to a peaceful evening ambiance.
> ```
>
> **Q6:** The authors use UniGen-DPO as a self-verifier to produce training data for test-time scaling. The model can generate low-quality or erroneous data, and it seems that no human correction has been applied to this data. How do the authors ensure data quality, and will low-quality data hamper model learning?
>
> **A6:** We kindly remind that when labeling training data, we used a much stronger pseudo labeler Qwen2.5VL-7B, not UniGen-DPO.
> We empirically find that the data quality is very high. To show this qualitatively, we randomly selected around 700 examples and ask human to do the annotation.
> Then, we compare the model labeled results and human labeled ones. We find that only 77 out of 692 (11%) samples have different results.
> This indicates that our model labeled data has very good quality.
>
> **Q7:** During training, UniGen first undergoes DPO and then the test-time scaling training. However, it seems that the model could first undergo the test-time scaling stage and then DPO, since training with CoT-V is simply supervised fine-tuning. This also makes sense to first foster the model’s CoT-based reasoning ability and then optimize its preference. What is the reason behind choosing the original order? Will this lead to better model performance? I recommend the authors provide more clarification on this point.
>
> **A7:** The reason we chose the original order is that the test-time scaling is the last stage in our pipeline and putting CoT-V tunning at the end would more likely ensure the model to follow the thinking instruction.
>
> We agree that it is interesting to try the other order. We flip the order of DPO and CoT-V in the following table. The results show that both strategies work and the original order has slight advantage on GenEval.
>
> |Method| Und. Avg   | GenEval | DPGbench |
> | ----- | ----- | ------- | -------- |
> |DPO->CoT-V| 62.7 | 0.78    | 85.19    |
> |CoT-V->DPO| 62.7 | 0.76  | 85.20    |
>
> **Q8:** typo: L257: stages -> stages.
>
> **A8:** Thanks. We will fix the typo.

---

> > ### Comment · Reviewer_WbDR · 2025-08-05
> >
> > Thanks for the author rebuttal. While some concerns are addressed, several issues remain.
> >
> > For A2, it appears that these experiences have only been validated during the development of UniGen, such as those for PT-1 and PT-2. However, these may not be generally applicable lessons. Have similar experiences been validated in other publications? Additionally, the authors may consider including this information in the main body of the paper.
> >
> > For A5, the authors primarily provide a prompt example without answering other sub-questions.

---

> > > ### Author Response · Authors · 2025-08-06
> > > **Response to Reviewer WbDR**
> > >
> > > Thank you for your thoughtful follow-up. Please see our responses below.
> > >
> > > ---
> > > **Further clarification of A2:**
> > > The training stages used in UniGen (PT-1, PT-2, SFT) are standard and commonly used in prior works such as Show-o [76], Janus [74], and Janus-Pro [8]. However, these works do not conduct a systematic exploration of strategies across different training stages. In contrast, we carefully designed and validated an effective training recipe for UniGen through comprehensive ablation studies. Our goal is to share these insights with the community in the hope that they may provide useful guidance for future research on unified MLLMs.
> > >
> > > While our lessons were derived during the development of UniGen, we believe some of the insights are implied in the design choices of prior works. Moreover, though previous works may not provide explicit ablations, our study offers a systematic exploration under the setting of unified MLLMs. For instance:
> > >
> > > - The importance of high-quality captions for image-text pretraining (L7) is supported by both understanding-only models (e.g., ShareGPT4V [1]) and generation models (e.g., PixArt-α [2]). To our knowledge, we are among the first to quantify this benefit in unified MLLMs.
> > > - Recent models like LLamaGen [3] and SimpleAR [4] discard the class-to-image warm-up stage and instead adopt direct text-to-image training. In UniGen, we proposes that the class-to-image pretraining used in [74,76,8] is not necessary (L4).
> > >
> > > We appreciate the suggestion to make these insights more visible in the main paper, and will discuss these lessons to the introduction and experiments sections.
> > >
> > > [1] ShareGPT4V: Improving Large Multi-modal Models with Better Captions
> > > [2] PixArt-α: Fast Training of Diffusion Transformer for Photorealistic Text-to-Image Synthesis
> > > [3] Autoregressive Model Beats Diffusion: Llama for Scalable Image Generation
> > > [4] SimpleAR: Pushing the Frontier of Autoregressive Visual Generation through Pretraining, SFT, and RL
> > >
> > > ---
> > > **Further clarification of A5:**
> > >
> > > **Q5.1:** "Since the Best-of-N strategy can also be combined with RV and achieve similar functionality, the superiority of CoT-V over RV remains unclear. "
> > >
> > > **A5.1:** First, we directly compare different verification strategies in Table 4 of the main paper. The results show that CoT-V outperforms RV by 3 points on GenEval, clearly demonstrating its advantage. Moreover, RV can not achieve effective score on DPGbench as it fails to decompose complex prompts using hand-crafted rules. Therefore, we respectfully argue that RV cannot achieve functionality comparable to CoT-V in general.
> > >
> > > **Q5.2:** "Can the authors give some examples of free-form or complex instructions to better illustrate RV’s limitation and give more details regarding CoT-V’s superiority?"
> > >
> > > **A5.2:** We further clarify the limitations of RV below. It is important to note that during inference, UniGen *only has access to the text prompt and the generated images, without any off-the-shelf visual questions*. To perform self-verification via fine-grained VQA, we realize that regular expression matching is capable of converting simple prompts from GenEval into reasonable visual questions. We refer to this verification method, which relies on manually crafted rules, as RV. For instance:
> > >
> > > **Example-1**
> > > - Category: counting
> > > - Prompt: a photo of [num] [object]
> > >     - e.g., a photo of two ovens
> > > - Question: Are there [object]?; Are there [num] [object]?
> > >     - e.g., Are there ovens? Are there two ovens?
> > >
> > > **Example-2**
> > > - Category: color attributes
> > > - Prompt: a photo of a [color1] [object1] and a [color2] [object2]
> > >     - e.g., a photo of a red dog and a blue cake
> > > - Question: Is there a [object1] ? Is the [object1] [color1]? Is there a [object2]? Is the [object2] [color2]?
> > >     - e.g., Is there a dog? Is the dog red? Is there a cake? Is the cake blue?
> > >
> > > However, for free-form prompts from DPGbench, e.g., the example provided in **A5**, it is practically infeasible to design rule-based patterns capable of matching all objects, attributes, and relations. In contrast, CoT-V manages to handle complex prompts automatically, without the need for any external rules, as shown below:
> > >
> > > > \<think_start\> Are there three remotes? yes; Are the remotes aligned? yes; Are the remotes sleek? yes; Is there a picture frame? yes; Is there a coffee table? yes; Is the picture frame black? no; Is the picture frame enclosing a photograph? yes; Is the photograph monochrome? yes; Is there a lamp? yes; Is the coffee table made of mahogany? yes; Is the living room's furniture earth-toned? yes; Are the remotes next to the picture frame? yes; \<think_end\>
> > >
> > > To clarify, *we do not position RV as a generalizable or practically applicable method. Instead, we treat it as an ideal upper bound for self-verification through fine-grained VQA*. In consequence, the superior performance of CoT-V over RV discloses that the reasoning process can improve the effectiveness and reliability of self-critique.

---

> > > > ### Comment · Reviewer_WbDR · 2025-08-09
> > > >
> > > > Thanks for your responses. Most of my concerns have been addressed and I will maintain the rating of borderline accept.

---

### Official Review · Reviewer_XW9f · 2025-07-04

**Clarity:** 3
**Significance:** 2
**Originality:** 3
**Rating:** 3
**Confidence:** 4

**Summary:**

This paper proposes a unified model for multimodal understanding and generation. To improve image generation quality, the authors introduce a method called Chain-of-Thought Verification (COT-V) for use during the inference phase. The model first generates a set of candidate images, then utilizes its multimodal understanding capabilities to score each image and selects the one with the highest score as the final output.

**Questions:**

Please see the weaknesses.

**Ethical Concerns:**

["NO or VERY MINOR ethics concerns only"]

**Limitations:**

yes.

**Paper Formatting Concerns:**

Nothing.

**Quality:**

3

**Strengths And Weaknesses:**

Strengths:
- The paper is clearly written and easy to follow. The motivation is reasonable, and the method is simple and reproducible.
- Leveraging the chain-of-thought reasoning process of multimodal models to evaluate image quality is an interesting idea.

Weaknesses:
- The core contribution, Chain-of-Thought Verification, does not appear to emerge naturally from the model but instead relies on fine-tuning with specific datasets. These datasets are based on rule-based scoring, which may limit the potential of the proposed method.
- There is randomness in the model's output during the chain-of-thought scoring process. How can one ensure that the output format consistently matches the expected structure, such as the format shown in Line 187?

---

> ### Author Rebuttal · Authors · 2025-07-31
>
> **To Reviewer XW9f**
>
> We thank the reviewer for their valuable feedback. Please see our response below. We are looking forward to seeing you increase the score. Please let us know if there are any more concerns.
>
> **Q1:** The core contribution, Chain-of-Thought Verification, does not appear to emerge naturally from the model but instead relies on fine-tuning with specific datasets. These datasets are based on rule-based scoring, which may limit the potential of the proposed method.
>
> **A1:** We argue that the instruction following capability emerges by only finetuning on the CoT-V training set for 500 steps. The light finetuning implies the naturalness of CoT-V for the model.
> For the concern of the rule-based scoring, we argue that rule-based scoring could also be generalizable given the recent success of RL with verifiable reward functions (rule-based) [1].
> Besides UniGen, we also finetuned Show-o with our dataset and showed that the Show-o can also be significantly improved (see Table 5). This also suggests the generated datasets are general.
>
> [1] DeepSeek-R1: Incentivizing Reasoning Capability in LLMs via Reinforcement Learning
>
> **Q2:** There is randomness in the model's output during the chain-of-thought scoring process. How can one ensure that the output format consistently matches the expected structure, such as the format shown in Line 187?
>
> **A2:** We calculate the percentage of data that follows the expected format and observe that UniGen sucessfully followed the instruction of CoT-V for 100% samples on GenEval, and only failed with one image-prompt input (0.13% in total) on DPGBench. The numbers suggest that our outputs consistently follow the expected format.

---

> > ### Author Response · Authors · 2025-08-01
> >
> > Dear Reviewer XW9f,
> >
> > Thanks again for your valuable feedback. Do you think our responses resolve your concerns? Feel free to ask more questions if you have any. We will appreciate it if you could consider raising the score.

---

> > > ### Author Response · Authors · 2025-08-07
> > >
> > > Dear Reviewer XW9f,
> > >
> > > Thanks again for your helpful feedback. We would like to make sure your concerns have been fully addressed. Please let us know if there are any other questions we can answer.

---

> > > > ### Author Response · Authors · 2025-08-08
> > > >
> > > > Dear Reviewer XW9f,
> > > >
> > > > We appreciate your valuable feedback. We are looking forward to more discussion. Thank you!

---

### Official Review · Reviewer_wZHu · 2025-07-14

**Clarity:** 3
**Significance:** 2
**Originality:** 2
**Rating:** 4
**Confidence:** 5

**Summary:**

# Summary

---

This work introduces a novel 1.5B-parameter model that unifies both image understanding and generation within a single framework. The model is trained through a comprehensive pipeline, including two-stage pre-training, SFT, DPO, and an additional CoT-V post-training phase to enable inference-time scaling with CoT-V techniques. For model design, the authors employ MAGVIT-v2 as the image tokenizer and utilize a cosine masking schedule for image decoding following MaskGIT. The image understanding encoder uses the widely adopted SigLIP. Both the encoder and decoder are kept frozen throughout all training stages, with only the LLM and connector modules being trainable. Notably, during pre-training, the image tokens are also optimized on understanding tasks to enhance the semantics of the image tokens.

**Questions:**

# Questions (bonus)

---

- I am also curious about the scaling performance of UniGen. In your scaling experiments (if any), did you encounter any bottlenecks caused by the generation decoder? Specifically, as the model size increases, did you observe that the image generation ability tends to saturate? If so, is this likely due to limitations of the image tokenizer, especially the generation decoder?
- How do the authors recap the PT-1 stage data using Qwen2.5-VL-7B? What prompts did you use for this process?

**Ethical Concerns:**

["NO or VERY MINOR ethics concerns only"]

**Final Justification:**

The authors’ responses have addressed most of my concerns. The additional ablation studies further enhance the depth and impact of this work, and several of my questions have been clarified in the supplementary materials. The practices and recipes presented serve as valuable guidelines for future research. However, the novelty of this work remains unclear, as CoT-V is essentially a BoN technique, which has already been explored in many previous studies.

**Limitations:**

Yes.

**Paper Formatting Concerns:**

None.

**Quality:**

2

**Strengths And Weaknesses:**

# Strength

---

- The model demonstrates impressive performance despite its small 1.5B parameter size.
- The CoT-V inference-time scaling technique is both innovative and interesting.
- The implementation is solid, with comprehensive training stages that span from pre-training to preference learning.

# Weakness

---

1. **What are the differences between the PT-1 and PT-2 stages?** Section 3.2 does not clearly state the distinction between these two stages. The datasets, augmented datasets, and tasks (t2i, i2t, t) all appear to be the same. Further clarification on how PT-1 and PT-2 differ would be helpful.
2. **What is the key novelty of UniGen compared to prior work?** UniGen seems to combine multiple existing modeling designs. For example, its architecture is highly similar to Janus, with both featuring decoupled generation (MAGViT) and understanding (SigLIP) encoders, as well as similar training strategies. While some differences exist, such as the use of a cosine masking schedule for parallel visual decoding, multi-stage pretraining, and frozen encoders throughout all stages, there is a lack of ablation studies to assess the impact of these techniques. The explanations for improved performance (lines 240–241) do not sufficiently address differences compared to Janus, leaving the true reasons for UniGen’s superior performance underexplored.
3. **Regarding Table 2**: Does only UniGen utilize BoN samples for image generation? If so, this could make the comparison unfair, as such techniques like contrastive reranking [1] can significantly affect generation performance.
4. **Additional ablation studies are needed**: Ablations should cover not only different training stages (PT-1, PT-2, SFT, DPO, CoT-V post-training), but also the key novelties of this work. For example:
- The effect of the generation encoder and decoder (vs. diffusion-based or flow matching-based methods)
- The impact of masking-based parallel visual decoding vs. causal decoding [2, 3]
- Frozen vs. tunable visual encoders during SFT (for comparison with Janus)
- The effect of different tokenizers
- The effect of not using the generation encoder for understanding tasks during pre-training
- (At least) Comparisons among OV, RV, and CoT-V
- Among others.
5. **Lack of evaluation on text-only tasks**: The paper does not report results on text-only tasks. How do the authors draw conclusions regarding preserved text performance?

[1] Scaling Autoregressive Models for Content-Rich Text-to-Image Generation

[2] Making LLaMA SEE and Draw with SEED Tokenizer

[3] MIO: A Foundation Model on Multimodal Tokens

---

> ### Author Rebuttal · Authors · 2025-07-31
>
> **To Reviewer wZHu**
>
> Thanks for the valuable suggestions. We added comprehensive experiments/ablations to address your concerns. Please see the detailed responses below. We are looking forward to seeing you increase the score. Please let us know if there are any more concerns.
>
> **Q1:** What are the differences between the PT-1 and PT-2 stages? Section 3.2 does not clearly state the distinction between these two stages. The datasets, augmented datasets, and tasks (t2i, i2t, t) all appear to be the same. Further clarification on how PT-1 and PT-2 differ would be helpful.
>
> **A1:** The major difference of PT-1 and PT-2 is the datasets used in those stages (see Table 14 in supplementary materials).
> In PT-1, our goal is to learn generating basic visual concepts for warming up, so we train with ImageNet.
> In PT-2, we want to learn broader visual concepts, so we make the T2I datasets more diverse by adding CC3M, CC12M and SA1B.
> We will make this more clear in the final version.
>
> **Q2:** What is the key novelty of UniGen compared to prior work? UniGen seems to combine multiple existing modeling designs. For example, its architecture is highly similar to Janus, with both featuring decoupled generation (MAGViT) and understanding (SigLIP) encoders, as well as similar training strategies. While some differences exist, such as the use of a cosine masking schedule for parallel visual decoding, multi-stage pretraining, and frozen encoders throughout all stages, there is a lack of ablation studies to assess the impact of these techniques. The explanations for improved performance (lines 240–241) do not sufficiently address differences compared to Janus, leaving the true reasons for UniGen’s superior performance underexplored.
>
> **A2:** We would like to kindly clarify that the novelty of this work does not lies in building new architectures.
> We claim our contributions in the following: First, with only publicly accessible training data, we walk through the entire training cycle and transparently show how metrics of each benchmark moves across different stages.
> Hopefully, the comprehensive ablation would share some useful tips to the community for building future unified MLLMs.
> Second, UniGen is the first unified MLLM that explicitly shows the generation and understanding capabilities can collaborate together for boosting performance via our proposed CoT verification.
>
> **Q3:** Regarding Table 2: Does only UniGen utilize BoN samples for image generation? If so, this could make the comparison unfair, as such techniques like contrastive reranking [1] can significantly affect generation performance.
>
> **A3:** Our baselines do not use BoN. We softly argue that each method utilizes their own strengths for boosting their results.
> For example, Janus-Pro uses much more training data including internal high quality data. Similarly, test-time scaling via CoT-V (BoN) is one of our contributions that idenfity an effective way to further boost the generation performance for unified MLLMs.
> To futher reduce the concern, we compare UniGen (without BoN) to one of the most top-performing Unified MLLM (Janus-Pro) as well as the best text-to-image generation model (infinity) as follows.
> The results show that even without BoN, UniGen performs better than these strong baselines.
>
> | Model          | GenEval | DPG-Bench |
> | -------------- | ------- | --------- |
> | Infinity       | 0.73    | 83.46     |
> | Janus-Pro      | 0.73    | 82.63     |
> | UniGen-1.5B w/o BoN | 0.74    | 84.89     |
>
> **Q4:** Additional ablation studies are needed.
>
> **Q4.1:** The effect of the generation encoder and decoder (vs. diffusion-based or flow matching-based methods)
>
> **A4.1:** Thanks for the helpful suggestion.
> Incorporating diffusion/flow matching in UniGen requires a lot of infra changes that is challenging to accomplish within the rebuttal time.
> A concurrent work BLIP3o is based on diffusion model with flow matching. Comparison with BLIP3o may be helpful to understand the pros and cons of diffusion based models.
> We scale up our model and compare with BLIP3o in the following table. Note that BLIP3o utilized GPT4o labeled data to boost performance and we incoporate their dataset in SFT stage for fair comparison.
> The results show that our method performs significantly better on GenEval and DPG-Bench.
> One interesting finding is that our UniGen demonstrates strong advantage on DPG-Bench. Our 1.5B model significantly outperforms BLIP3o-8B by ~3.6 points. This may imply that our method has much better instruction following capability for generation, since most of the prompts in DPGBench are long/complicated with many fine-grained elements.
> | Model          | GenEval | DPG-Bench |
> | -------------- | ------- | --------- |
> | UniGen-1.5B | 0.78    | 85.19     |
> | BLIP3o-8B     | 0.84    | 81.6     |
> | UniGen-7B* | 0.90    | 86.38     |
>
> [1] BLIP3-o: A Family of Fully Open Unified Multimodal
> Models—Architecture, Training and Dataset
>
> **Q4.2:** The impact of masking-based parallel visual decoding vs. causal decoding.
>
> **A4.2:** To answer your question, we add experiments by replacing masked based decoding with the causal decoding. The comparison after SFT stage is shown as follows.
> As we can see, on average, the causal decoding baseline performs much worse on both understanding and generation benchmarks. This implies that full attention across vision tokens has advantages over the casual attention in our design.
>
> |Method| Und. Avg   | GenEval | DPGbench |
> | ----- | ----- | ------- | -------- |
> |mask decoding| 63.11 | 0.63    | 82.75    |
> |causual decoding| 61.94 | 0.53    | 80.99    |
>
> **Q4.3:** Frozen vs. tunable visual encoders during SFT (for comparison with Janus)
>
> **A4.3** We add an experiment by unfreezing the visual encoder during SFT and report the results as follows. The results demonstrate that unfreezing visual encoder shows on-par performance compared to the frozen one, except that there is notable 2% improvements on GenEval.
> Since freezing visual encoder is more cost friendly, we would suggest to go with the frozen design.
>
> |Visual Encoder| Und. Avg   | GenEval | DPGbench |
> | ----- | ----- | ------- | -------- |
> |Frozen| 63.11 | 0.63    | 82.75    |
> |Unfreeze| 63.16 | 0.65    | 82.71    |
>
> **Q4.4:** The effect of different tokenizers
>
> **A4.4:** We add an experiment by replacing the default tokenizer (MAGViTv2) with VQ-16 from LLamaGen[2]. The comparison after SFT stage is shown as follows.
> We can see that VQ-16 leads to a similar performance as MAGViTv2, verifying the generalizability of our training framework.
>
> | Tokenizer |GenEval | DPGbench |
> | ----- | ------- | -------- |
> | MAGViTv2| 0.63    | 82.75    |
> | VQ-16 |  0.62  |  82.93  |
>
> [2] Autoregressive Model Beats Diffusion: Llama for Scalable Image Generation
>
> **Q4.5:** The effect of not using the generation encoder for understanding tasks during pre-training
>
> **A4.5:** We have shown in Table 8 (supplementary materials) that completely removing understanding tasks during pretraining will negatively impact understanding performance.
> Moreover, we add one more experiment that uses a separate SigLIP encoder for understanding during pretraining. The results after SFT are shown as follows. The results indicate that using a separate SigLIP encoder during pretraining will significantly improve understanding performance while sacrificing generation performance.
>
> | Enc. for Und. | Und. Avg   | GenEval | DPGbench |
> | ----- | ----- | ------- | -------- |
> | MAGViTv2|  63.11| 0.63    | 82.75    |
> | SigLIP | 65.02 |  0.61|    79.82  |
>
> **Q4.6:** (At least) Comparisons among OV, RV, and CoT-V
>
> **A4.6:** For comparisons among the three verification methods, we have provided detailed analysis in section 4.3.2 and Table 4.
>
> **Q4.7:** Lack of evaluation on text-only tasks: The paper does not report results on text-only tasks. How do the authors draw conclusions regarding preserved text performance?
>
> **A4.7:** Thanks for the suggestion. It would be optimal, but not trivial to have no regression on text-only task.
> We follow prior works (Janus-pro, Show-o) that do not focus on text-only tasks. We believe it will be an interesting future work.
>
> **Q5:** I am also curious about the scaling performance of UniGen. In your scaling experiments (if any), did you encounter any bottlenecks caused by the generation decoder? Specifically, as the model size increases, did you observe that the image generation ability tends to saturate? If so, is this likely due to limitations of the image tokenizer, especially the generation decoder?
>
> **A5:** We add an experiment that scales up the model size of UniGen to 7B under our default training pipeline and data mixtures.
> Note that due to the limited time, we use the DPO and CoT-V data generated by UniGen-1.5B for training UniGen-7B. We expect more increase if we use UniGen-7B for creating its own data.
> Nevertheless, we observe increase on both understanding and generation benchmarks. We believe better image tokenizer leads to better generation quality.
> Improving tokenizer is one promising future work in our roadmap.
>
> | Model          | Und Avg.|GenEval | DPG-Bench |
> | -------------- | ------- |------- | --------- |
> | UniGen-1.5B | 62.66| 0.78    | 85.19     |
> | UniGen-7B | 63.78| 0.79    | 85.79     |
>
> **Q6:** How do the authors recap the PT-1 stage data using Qwen2.5-VL-7B? What prompts did you use for this process?
>
> **A6:** We have provided the detailed prompts in the supplementary materials. Please refer to E03.

---

> > ### Author Response · Authors · 2025-08-01
> >
> > Dear Reviewer wZHu,
> >
> > Thanks again for your helpful comments. We really appreciate it if you could consider raising the score. Feel free to to let us know if you have more questions.

---

> > > ### Author Response · Authors · 2025-08-07
> > >
> > > Dear Reviewer wZHu,
> > >
> > > Thanks again for your valuable comments. Please let us know if our response resolved your concerns. We are looking forward to more discussions.

---

> ### Comment · Reviewer_wZHu · 2025-08-08
> **Response**
>
> Thank you for your rebuttal. Most of my concerns have been addressed. Accordingly, I have raised my score to a positive one.

---

> > ### Author Response · Authors · 2025-08-08
> >
> > Thank you for raising your score!

---

### Author Response · Authors · 2025-08-05

Dear Reviewers,

Thank you again for your time and thoughtful feedback.

We have carefully addressed all comments in our rebuttal through providing detailed clarifications, conducting further ablation studies, and scaling up the model size. We hope our responses have resolved your concerns and helped clearly convey the contributions of UniGen.

If you find the rebuttal satisfactory, we would be grateful if you could consider raising your score. If any points remain unclear, we would sincerely appreciate the opportunity for further discussion.

---

### Note · Authors · 2025-08-13

We thank the AC for managing the review process and the reviewers for their constructive feedback and engagement. We are encouraged by the recognition across reviewers:

- **Reviewer wZHu** described CoT-V as “both innovative and interesting” and noted UniGen’s impressive performance despite its small 1.5B parameter size.
- **Reviewer XW9f** described CoT-V as “an interesting idea” with a well-motivated design.
- **Reviewer WbDR** praised the novelty and generalizability of CoT-V and emphasized that our constructed preference dataset would be a “valuable contribution to the community”.
- **Reviewer CnR8** acknowledged our “clear and systematic methodology” and noted that UniGen “achieves notable improvements over prior unified MLLMs”.

During the rebuttal, we conducted extensive ablation studies, clarifications, and scaling experiments to address all raised concerns, actively responding to additional feedback and engaging in discussions with the reviewers. Our efforts included:
1. **Expanding Ablations**: We conducted additional ablation studies across training stages and key design choices, including decoding strategies, unfreezing visual encoder, tokenizer choices, and the sequence of DPO and CoT-V training.
2. **Clarifying Novelty**: We claimed the contribution of UniGen in the following:
    - Using only publicly available data, we walk through the full training cycle and transparently show how benchmark metrics change across stages. We hope the comprehensive ablations would provide useful tips to the community for building future unified MLLMs.
    - UniGen is the first unified MLLM that explicitly shows the understanding capability can help to boost the generation performance via our proposed CoT verification.
3. **Summarizing Lessons**: We summarized 8 meaningful lessons drawn from different training stages of UniGen.
4. **Scaling Up UniGen**: We scaled model size to 7B and observed consistent gains. We found that the scaled-up UniGen demonstrates strong advantage on DPG-Bench compared to BLIP3-o, a concurrent work using the flow-matching decoder.
5. **Analyzing Costs of CoT-V**:  We analyzed the test-time overhead of CoT-V, and displayed that compared with 7b model, CoT-V on UniGen-1.5b can effectively narrow the performance gap with similar inference cost.

We once again thank the AC and all reviewers for their time and feedback. We will ensure that all suggested revisions and clarifications are incorporated into the final version.

---

### Note · Authors · 2025-08-13

We thank the AC for managing the review process and the reviewers for their constructive feedback and engagement. We are encouraged by the recognition across reviewers:

- **Reviewer wZHu** described CoT-V as “both innovative and interesting” and noted UniGen’s impressive performance despite its small 1.5B parameter size.
- **Reviewer XW9f** described CoT-V as “an interesting idea” with a well-motivated design.
- **Reviewer WbDR** praised the novelty and generalizability of CoT-V and emphasized that our constructed preference dataset would be a “valuable contribution to the community”.
- **Reviewer CnR8** acknowledged our “clear and systematic methodology” and noted that UniGen “achieves notable improvements over prior unified MLLMs”.

During the rebuttal, we conducted extensive ablation studies, clarifications, and scaling experiments to address all raised concerns, actively responding to additional feedback and engaging in discussions with the reviewers. Our efforts included:
1. **Expanding Ablations**: We conducted additional ablation studies across training stages and key design choices, including decoding strategies, unfreezing visual encoder, tokenizer choices, and the sequence of DPO and CoT-V training.
2. **Clarifying Novelty**: We claimed the contribution of UniGen in the following:
    - Using only publicly available data, we walk through the full training cycle and transparently show how benchmark metrics change across stages. We hope the comprehensive ablations would provide useful tips to the community for building future unified MLLMs.
    - UniGen is the first unified MLLM that explicitly shows the understanding capability can help to boost the generation performance via our proposed CoT verification.
3. **Summarizing Lessons**: We summarized 8 meaningful lessons drawn from different training stages of UniGen.
4. **Scaling Up UniGen**: We scaled model size to 7B and observed consistent gains. We found that the scaled-up UniGen demonstrates strong advantage on DPG-Bench compared to BLIP3-o, a concurrent work using the flow-matching decoder.
5. **Analyzing Costs of CoT-V**:  We analyzed the test-time overhead of CoT-V, and displayed that compared with 7b model, CoT-V on UniGen-1.5b can effectively narrow the performance gap with similar inference cost.

We once again thank the AC and all reviewers for their time and feedback. We will ensure that all suggested revisions and clarifications are incorporated into the final version.

---

### Decision · Program_Chairs · 2025-09-17

**Decision:**

Accept (poster)

**Comment:**

This paper contributes UniGen, a 1.5B sized VLM trained for both image understanding and generation. The model design is based on existing architectures and components, such as MAGViT-v2 for tokenization, MaskGIT schedule for generation, and SigLIP for understanding. At its core sits an autoregressive LLM that operates on tokens and is the main focus of training. The paper also contributes Chain-of-Thought Verification (CoT-V,) an inference-time scaling technique that leverages the model's ability to act as a critic to improve its generations.

Overall, the reviewers are satisfied with the contribution, highlighting its significance, and value for other researchers to continue from.
Reviewers see the value both in showcasing how to train a large VLM and in CoT-V, which makes progress on an important problem.
A reviewer concern was the lack of novelty over prior approaches, such as Janus, which is insufficiently compared to. In response, the authors argue that their contribution lies in demonstrating how to walk through the entire training process using publicly available data, and CoT-V, which shows how to leverage generation and understanding capabilities in a single model. Another concern is that for CoT-V to work, targeted finetuning is needed (relying on a rule-based verifier for scoring), somewhat limiting the naturalness of the approach. It is argued that this later might be resolved via RL, and the datasets used are general enough to improve another model in this regard as well. Other issues raised by the reviewers were sufficiently addressed by the authors in their response.

From the reviewer scores, this paper is somewhat borderline. One reviewer, recommending LR did not participate in the rebuttal, which was taken into account. Other reviewers recommend LA, though none of the reviewers is willing to champion the work. In the AC-reviewer discussion a concern was raised that CoT-V is too similar to relevant prior work on BoN sampling for pure image generation. Overall my assessment is that there is enough of a contribution here and I will mark as accept for now.